



# Review of regional Antarctic snow accumulation over the past 1000 years

Elizabeth R. Thomas[1], J. Melchior van Wessem[2], Jason Roberts[3,4], Elisabeth Isaksson[5], Elisabeth Schlosser[6,7], TJ Fudge[8],

Paul Vallelonga[9], Brooke Medley[10], Jan Lenaerts[2,] Nancy Bertler[11, 12], Michiel R. van den Broeke[2], Daniel A. Dixon[13]
Massimo Frezzotti[14] Barbara Stenni[15, 16], Mark Curran[3] Alexey A. Ekaykin[17,18]

[1] British Antarctic Survey, Cambridge, UK CB3 0ET

[2] Institute for Marine and Atmospheric research Utrecht (IMAU), Utrecht University, Utrecht, The Netherlands

[3] Australian Antarctic Division, Tasmania 7050, Australia

[4] Antarctic Climate & Ecosystems Cooperative Research Centre, University of Tasmania, Hobart, 7001, Australia

[5] Norwegian Polar Institute, 9296 Tromsø, Norway

[6] Austrian Polar Research Institute, Vienna, Austria

[7] Inst. of Atmospheric and Cryospheric Sciences, Univ. of Innsbruck, Innsbruck, Austria

[7] University of Washington, Seattle, USA

[8] Centre for Ice and Climate, Niels Bohr Institute, University of Copenhagen, Copenhagen, Denmark.

[9] Cryospheric Sciences Laboratory, NASA Goddard Space Flight Center, Greenbelt, Maryland, USA

[10] Antarctic Research Centre, Victoria University, Wellington 6012, New Zealand

[11] National Ice Core Research Laboratory, GNS Science, Lower Hutt 5040, New Zealand

[12] Climate Change Institute, University of Maine, Orono, Maine 04469, USA

[14] ENEA, Agenzia Nazionale per le nuove tecnologie, l'energia e lo sviluppo sostenibile, Rome, Italy

[15] Department of Environmental Sciences, Informatics and Statistics, Ca' Foscari University of Venice, Italy

[16] Institute for the Dynamics of Environmental Processes, CNR, Venice, Italy

[17] Climate and Environmental Research Laboratory, Arctic and Antarctic Research Institute, St. Petersburg, 199397, Russia

[18] Institute of Earth Sciences, Saint Petersburg State University, St. Petersburg, 199178, Russia

*Correspondence to:* Elizabeth. R Thomas (lith@bas.ac.uk)

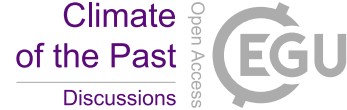

**Abstract.** Here we review Antarctic snow accumulation variability, at the regional scale, over the past 1000 years. A total of 80 ice core snow accumulation records were gathered, as part of a community led project coordinated by the PAGES Antarctica 2k working group. The ice cores were assigned to seven geographical regions, separating the high accumulation coastal zones below 2000m elevation from the dry central Antarctic Plateau. The regional composites of annual snow accumulation were evaluated against modelled surface mass balance (SMB) from RACMO2.4 and precipitation from ERA-
interim reanalysis. With the exception of the Weddell Sea coast, the low-elevation composites capture the regional precipitation and SMB variability. The central Antarctic sites lack coherency and are either not representing regional precipitation or indicate the models inability to capture relevant precipitation processes in the cold, dry central plateau. The drivers of precipitation are reviewed for each region and the temporal variability and trends evaluated over the past 100, 200 and 1000 years. Our study suggests an overall increase in SMB across the grounded Antarctic ice sheet of ~44 GT since
1800 AD, with the largest (area-weighted) contribution from the Antarctic Peninsula (AP). Only four ice core records cover the full 1000 years and suggest a decrease in snow accumulation during this period. However, our study emphasizes the importance of low elevation coastal zones (especially AP and DML), which have been underrepresented in previous investigations of temporal snow accumulation.



## 1. Introduction

The Antarctic Ice Sheet (AIS) is the largest reservoir of fresh water on the planet and has the potential to raise global sea level by about 58.3 m if melted completely (IPCC, 2013). Even small changes in its volume could have significant impacts,

not just on global mean sea level, but also on the wider hydrological cycle, atmospheric circulation, sea surface temperature, ocean salinity, and thermohaline circulation. The mass balance of the AIS constitutes the difference between mass gains, mainly by snow accumulation, and mass losses, mainly by ice flow over the grounding line. Ice sheet mass balance is currently estimated in three ways: 1) Ice sheet volume change is calculated using repeated surface elevation measurements from radar/laser altimeters on airborne/spaceborne platforms, followed by a conversion from volume change to mass change

using a model for firn density. 2) Ice sheet mass changes can be directly measured using the Gravity Recovery And Climate Experiment (GRACE) satellite system. 3) Surface mass balance (SMB) and solid ice discharge can be individually estimated and subtracted (Rignot et al., 2011). These three techniques have significantly advanced our understanding of contemporary AIS mass balance, with growing evidence of increased mass loss over recent decades (Velicogna and Wahr, 2006; Allison et al., 2009; Chen et al., 2009; Rignot et al., 2011; Shepherd et al., 2012). However, the uncertainties in these assessments may

be as high as 75% (Shepherd et al., 2012), and another study even suggested a positive trend over the same period (Zwally et al., 2015).

The discrepancies and uncertainties may in part reflect the large interannual (Wouters et al., 2013) and spatial (Anschuetz et al., 2006) variability in snowfall and hence SMB, but also the difficulty with which SMB is measured and the complexities

of the data interpretation. AIS SMB is the net result of multiple processes such as surface mass gains from snowfall and deposition, and losses from snow sublimation and wind erosion. Mass losses from meltwater runoff are small in Antarctica, except for some parts of the Antarctic Peninsula. Multiple in situ SMB observations can be combined to produce an SMB map for selected ice sheet regions (Rotschky et al., 2007) or for the entire AIS (Favier et al., 2013). Alternatively, SMB can be simulated by (regional) atmospheric climate models, such as the Regional Atmospheric Climate Model (RACMO)

version 2.3 (Van Wessem et al., 2014) and various reanalysis products (such as ERA-Interim, Dee et al., 2001) or JRA55 (Kobayashi et al., 2015). Estimates of SMB can be improved by combining methods following e.g. the approach of Arthern et al. (2006), interpolating field measurements with remotely sensed data as a background field, or Van de Berg et al. (2006), who fitted output from a regional model to in situ observations. The resulting values of SMB averaged over the grounded AIS range from 143 kg m$^{-2}$ yr$^{-1}$ (Arthern et al., 2006) to 168 kg m$^{-2}$ yr$^{-1}$ (van de Berg et al., 2006). Several studies have

evaluated modelled SMB with in situ observations across Antarctica (Thomas and Bracegirdle, 2009; 2015; Agosta et al., 2012; Sinisalo et al., 2013; Medley et al., 2013 and Wang et al., 2015). Finally, the resulting maps of SMB can be combined with estimates of dynamical mass loss to estimate regional or continental ice sheet mass balance.




An increase in Antarctic SMB is expected in a warmer climate, as a result of increased precipitation when atmospheric moisture content increases (Krinner et al., 2007; Agosta et al., 2013; Ligtenberg et al., 2013; Frieler et al., 2015), with climate models on average predicting a 7.4 %/°C precipitation increase per degree of atmospheric warming (Palerme et al., 2017). This potentially results in a mitigation of sea level rise in the future (Krinner et al. 2007; Agosta et al., 2013) ranging

from 25 to 85 mm during the 21[st] century, depending on the climate scenario (Palerme et al., 2017). However, almost all of the models in the Fifth Climate Model Intercomparison Project (CMIP5) overestimate Antarctic precipitation (Palerme et al., 2017).

In order to better constrain predictions of future contributions to global sea level, it is therefore of vital importance to gain a

thorough understanding of past and present changes in SMB, and its relationship with the climate system. Whereas the methods discussed above are invaluable in determining contemporary SMB and its spatial variability, only ice core records have the ability to investigate past SMB beyond the instrumental and satellite period. Previous studies have evaluated ice core records to reveal an insignificant change in Antarctic accumulation rates since the 1950s (Monaghan et al., 2006a; 2006b) and Frezzotti et al. (2013) extended this analysis to conclude that the current SMB is not exceptionally high in the

context of the past 800 years.

However, in recent years the number of ice core accumulation records have increased, revealing large differences in the spatial pattern of snow accumulation and trends. Ice core records from the Antarctic Peninsula (AP) show dramatic increases in snowfall during the 20[th] century (Thomas et al., 2008, Goodwin et al., 2015), which appear unusual in the context of the

past ~300 years (Thomas et al., 2015). In Dronning Maud Land (DML), similarly large increases in precipitation have been observed in an ice core from a coastal ice rise (Philippe et al., 2016), consistent with remotely sensed estimates of recent mass gain (Shepherd et al., 2012). However, other coastal cores in DML exhibit a decrease in SMB in recent decades (Altnau et al., 2015; Schlosser et al., 2014; Sinisalo et al., 2013), despite evidence of several high-accumulation years since 2009 related to a persistent atmospheric blocking flow pattern (Boening et al., 2012; Lenaerts et al., 2013; Schlosser et al.,

2016). In Wilkes Land, the Law Dome ice core record reveals similarly elevated accumulation during the most recent 30 years, but also periods of elevated snowfall ~1600 and ~1200 years ago (Roberts et al., 2015). The growing evidence for large regional differences in snow accumulation trends demonstrates the need for a regionally focused study of snow accumulation beyond the instrumental period.

Here we review the available Antarctic ice core snow accumulation records as part of the PAGES Antarctica 2k community effort. The aim is to improve regional estimates of SMB variability using well constrained and quality checked snow accumulation records from ice cores. We will assess the regional representativeness of ice core snow accumulation composites, and the spatial pattern of variability, by utilising precipitation data derived from reanalysis products and SMB



from the RACMO2.4 regional climate model. We review the dominant atmospheric drivers of regional SMB and evaluate the changes over centennial timescales, and where possible place these changes in the context of the past 1000 years.

## 2. Data and Methods

### 2.1. Snow accumulation from ice cores

Ice cores have the potential to record the amount of snow accumulation at a specific location over a range of timescales. Barring post-depositional processes, the recorded snow accumulation is the net result of solid precipitation, sublimation, wind erosion/deposition and meltwater runoff. Integrated over the AIS the contributions made by sublimation/deposition, wind redistribution, rainfall and meltwater runoff are relatively small (van Wessem et al., 2014) and therefore the dominant component of Antarctic SMB is solid precipitation. Locally, however, drifting snow erosion/deposition and sublimation may play an important role, especially in regions of strong katabatic flow (Lenaerts and Van den Broeke, 2012).

### 2.2. Calculating snow accumulation and correcting for thinning and flow

Estimates of snow accumulation are based on the physical distance between suitable age markers within the ice core, corrected for firn density and the integrated influence of the vertical strain rate profile (so-called "layer thinning"). Depending on the time-scale of interest, age markers may include bulk changes in isotopic composition reflecting glacial cycles, volcanic eruptions for decadal-to-millennial time-scales to seasonal variations in stable water isotopes and chemical species including sea-salts, hydrogen peroxide, radio-isotopes and biological controlled compounds (Dansgaard and Johnsen 1969). Once suitable age markers have been identified, the effects of firn density can be corrected for by convolving the physical depth in the ice core (z') by the density as a function of depth ($\rho(z')$) as a fraction of a reference density ($\rho_{ref}$), to give an equivalent depth (z),

$$z = \frac{1}{\rho_{ref}} \int_{surface}^{z'} \rho(y)dy$$

Typically the reference density is either the density of glacial ice (resulting in "ice equivalent" measurements) or taken as 1000 kg m$^{-3}$ to give "water equivalent" (w.e) results.

Finally, due to the differential vertical velocity with depth (the vertical strain rate), the distance between layers that were equally spaced at the surface will decrease with depth. If the profile of vertical strain rate with depth is known, and is assumed to be invariant over time, then it can be corrected for. However, typically the vertical stain rate is unknown. In the absence of any other data, the vertical strain rate profile can be approximated as a constant (Nye, 1963), which in general is applicable in the upper portion of the ice sheet. Further refinement to this approximation was suggested by Dansgaard and

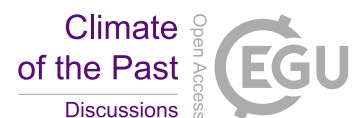

Johnsen (1969) who proposed a piecewise linear vertical strain rate profile, constant in the upper portion of the ice sheet and below that decreasing linearly to zero at the base of the ice-sheet, or a non-zero value in the presence of basal melt.

Roberts et al., (2015) show that a more realistic vertical strain rate profile, based on the power law distribution of horizontal velocity of Lliboutry (1979), may provide a better estimate of strain rate even in the upper portion of the ice sheet. Additionally, recently deployed ground based phase sensitive ice penetrating radar systems have demonstrated the ability to measure the vertical strain rate profile (Nicholls et al., 2015).

### 2.3. Antarctica 2k snow accumulation database

The criteria for the Antarctica 2k database was that all ice core derived snow accumulation records must be published and peer reviewed, have an annual resolution and cover at least the reference time period of 1960-1990. All records are corrected for thinning (following a suitable method as described above) and presented in m w.e.

From a total of 144 records submitted to the Antarctica 2k database, 80 records were eligible for this study (Table 1), building upon previous SMB compilation studies by Favier et al., (2013) and Frezzotti et al., (2013). Of the 80 records selected, 41 records extend over the past 200 years, and therefore this period has been selected as a focus for our reconstruction. All spatial correlations, plots and trends presented during this period (with the exception of the Weddell Sea Coast (WS)) are based on multiple records in each region, to avoid introduced errors associated with the switch to single sites. Prior to 1800 the number of records drops dramatically, with only eight records covering the past 500 years, four records covering the past 1000 years and only the Law Dome, RICE and WAIS ice cores covering the full 2000-year period. Therefore it was decided to limit our analysis to the past 1000 years.

### 2.4. Modelled surface mass balance

In this study we utilize the precipitation fields from the European Centre for Medium-Range Weather Forecasts (ECMWF) Interim Re-Analysis (ERA) (1979-onwards) (Simmons et al., 2007; Dee et al, 2011). It replaces the previous ERA-40 reanalysis, with improved model physics and observational data supplemented by ECMWF's operational archives, providing a better representation of the hydrological cycle in the high Southern Hemisphere latitudes than earlier reanalyses (Bromwich et al., 2011; Bracegirdle and Marshall, 2012). A recent evaluation of 3265 multiyear averaged in situ observations concluded that ERA-interim exhibits the highest performance of capturing interannual variability on observed Antarctic precipitation out of the available reanalysis products (Wang et al., 2016).



In addition, we use the latest version of RACMO version 2.4 (RACMO2.4), which succeeds RACMO2.3 (Van Wessem et al., 2014) and combines the hydrostatic dynamics of the High Resolution Limited Area Model (HIRLAM) with the physics package of the ECMWF Integrated Forecast System. RACMO2.4 has been specifically adapted for use over glaciated regions, as it is coupled to a multilayer snow model (Ettema et al., 2010) that calculates processes in the firn, such as grain

size growth, firn compaction, meltwater percolation/retention and meltwater runoff. As it also includes a drifting snow routine that calculates sublimation and divergence/convergence of blowing snow (Lenaerts et al 2012), RACMO2.4 explicitly calculates all SMB components over the AIS.

ERA-Interim reanalysis data (Dee et al., 2011) force the model at its lateral boundaries and prescribe sea surface temperature

and sea-ice cover for the period 1979-2015. The relaxation zone, where the forcing is applied to RACMO2.4, is located over the oceans, such that the model is able to evolve freely over the continent. An important update in RACMO2.4, with respect to RACMO2.3, is the addition of upper-air relaxation at the two top model layers as described in (Van de Berg and Medley, 2016.), which improves the interannual variability of the modelled SMB, which now better matches annually resolved ice core records.

## 2.5.   Defining the regions

One of the goals of this paper is to investigate the regional differences in snow accumulation, reducing the bias introduced in previous continental reconstructions, where regions of high data density overpower the signal in the data sparse, low elevation and coastal zones. Therefore, the Antarctic continent is divided into seven geographical regions (Fig. 1) each with

distinct climates. The East Antarctic Plateau (EAP) corresponds to locations in East Antarctica above 2000 m elevation, allowing separation of ice cores with a continental signal from those with a marine signature. The coastal regions of East Antarctica were separated into DML, defined as the areas between 15° W and 70° E, Wilkes Land Coast (WL) (70° E -150° E), and Victoria Land (VL) (150° E -170° E). The Weddell Sea Coast (WS) covers the Ronne-Filchner ice shelf and into Coats Land (60-15° W). Note the difference from the common usage of the geographical names, DML and WL in our

definition that refer only to the coastal regions, whereas the higher altitude regions belong to EAP. In this study the AP is treated separately from the West Antarctic Ice sheet (WAIS), with a division at 88° W.

## 2.6.   Methods for composite time series

The ice cores in the Antarctica 2k database were first separated into geographical regions (Fig. 1) and standardised relative to

the reference period 1960-1990. Based on the defined boundaries, the largest density of data is the EAP (accounting for 45 % of the records), nevertheless the spatial distribution of data is often poor. There are inadequate records in high accumulation coastal and slope areas and in the vast polar plateau, where snow accumulation is lower than 70 mm w.e. yr$^{-1}$ and seasonally deposited chemical or physical signals are frequently erased or changed by the action of the near- surface wind (Eisen et al.,





2008). To avoid biases introduced because of high data density, especially relevant in areas such as EAP and WAIS, records from the same grid box (chosen with a size of 2° latitude vs 10° longitude) were averaged together. The standardized records were then averaged together to form the standardized regional composites.

**3.  Results and discussion**

   **3.1.  Representative of regional SMB?**

Due to small-scale variations in surface slope and roughness, the snow accumulation at an ice core site may vary somewhat from the local spatial mean. Before we investigate the temporal changes in the records we first establish how representative each composite is of regional SMB, by testing against modelled SMB from RACMO2.4 (Fig. 2) and precipitation from

ERA-interim (Fig. 3) (direct SMB is not available from the reanalysis product). Areas of high correlation (red) indicate that a large fraction of snow accumulation variability in the composite time series is explained by the modelled SMB or precipitation variability during the period 1979-2010. The cut-off of 2010 was chosen as very few records extend beyond this.

Comparison with both RACMO2.4 SMB and ERA-interim precipitation suggests that the composite snow accumulation records from AP, WAIS, WL, DML, and VL (Fig. 2 and 3) are indeed representative of regional accumulation.  These records capture ~25-40 % of the variance in their respective regions and may be used to investigate changes in SMB beyond the instrumental period.  It is important to note that this comparison covers a relatively short time period (1979-2010) and a period of known climate forcing. In the Pacific sector the calibration period falls within a predominantly positive phase of

the Interdecadal Pacific Oscillation (IPO), which is known to influence the atmospheric circulation transporting moisture to Law Dome (Vance et al., 2015).  The Southern Annular Mode (SAM), the dominant mode of atmospheric variability in the southern hemisphere, is also predominantly in its positive phase during this period, which has been demonstrated to influence precipitation especially in AP (Thomas et al., 2008; Abram et al., 2014) and WAIS (Fogt et al., 2012; Raphael et al., 2016). Therefore, some care must be taken when extrapolating results beyond the instrumental period, when the same

climate forcings may be different.

Despite the good agreement in most regions, the opposite is true for the records from EAP and WS. WS represents a single ice core and therefore we have been unable to reduce the signal to noise ratio achieved in the regions with multiple records spread over a large area. The poor relationship between snow accumulation and precipitation at this site may be a result of

post depositional changes at the coastal dome or reflect the models inability to capture precipitation and SMB variability at this site. In addition, there is a relatively small period of overlap with the modelled and reanalysis datasets (1979-1992).





The EAP has many records that cover the full calibration period (1979-2010), but they are restricted in the area of DML plateau around Kohnen station, South Pole and Talos Dome, while the inner part is represented only by the Vostok site. As such, the EAP composite is poorly related to SMB (Fig. 2a and 3a), but reducing the EAP area and producing smaller "sub-regions" is no improvement. One possible explination is that the models do not take into account precipitation processes

such as diamond dust, likely having a relatively large influence on precipitation variability in these extremely cold areas (van de Berg et al., 2005; Mahesh at al., 2001). Additionally, large glaze/dune fields cover a large percentage (more than 500,000 km$^2$) of the EAP (Fahnestock et al., 2000) and these have been shown to significantly confound accumulation measurements in ice cores (Dixon et al., 2013). A combination of large topographical gradients and strong katabatic winds provides challenges for models in the grounding line area and this is where the largest differences appear between field data and the

modeled SMB (e.g. Sinisalo et al., 2014). Large areas along the margins of the EAP are characterized by steep slopes and are thus often suffering from challenges in describing the physical processes related to SMB in the lower resolution model domains. For completeness we are including EAP and WS composites in this study, however, caution should be used when interpreting the climatological drivers and temporal changes in these regions.

The standardized regional composites have been averaged together to form a West Antarctic ("WEST" combines AP, WAIS and VL), East Antarctic ("EAST" combines WS, DML, WL and EAP) and continental ("ALL" combines all records) composite, adopting a similar approach to previous SMB studies focusing on continental reconstructions (eg Monaghan et al., 2006a; Frezzotti et al., 2013). Evaluation with RACMO2.4 (Fig. 2) and ERA-interim (Fig. 3), confirm that continental style reconstructions reduce the representativeness of SMB, and are susceptible to bias from high sampled and high

accumulation areas. Such that WEST resembles AP, EAST resembles WL and ALL is a combination of these two regions, but with the area of significance across the whole continent significantly reduced. For this reason, our study will focus on regional SMB and not attempt to produce a continental reconstruction.

Spatial correlation plots with SMB from RACMO2.4 (Fig. 2) and precipitation from ERA-interim (Fig. 3) highlight some

interesting relationships. Significant positive correlations with precipitation in AP are mirrored by negative correlations in WAIS, especially the region around Marie Byrd Land (Fig. 2d and 3d). The relationship is reversed for the WAIS composite, which reveals positive correlations with precipitation in continental WAIS but negative correlations in the northern AP (Fig. 2e and 3e). This pattern likely reflects the influence of the Amundsen Sea Low (ASL) and its association with the Antarctic Dipole and ENSO (Yuan & Martinson, 2001). This persistent area of low pressure, the result of the frequency and intensity

of individual cyclones (Baines and Fraedrich, 1989; Turner et al., 2013), is known to affect the climatic conditions on the AP and WAIS (Hoskings et al., 2013; Thomas et al., 2015; Raphael et al., 2016).

In East Antarctica, positive correlations exist between WL and DML due to similar synoptic regimes and the influence of changes in the general atmospheric circulation, such as the Southern Annular Mode (SAM, e.g. Marshall, 2003) or zonal



wave number 3 (ZW3) index (Raphael, 2007). We explore the drivers of regional SMB further, using the available meteorological data from ERA-interim together with satellite observations of sea ice conditions.

### 3.2. Drivers of regional precipitation

#### 3.2.1. East Antarctic Plateau (EAP)

Accumulation in EAP is generally extremely low (<50mm/a on the high plateau) and exhibits high interannual variability. The temporally prevailing type of precipitation is diamond dust, consisting of very fine needles or platelets formed when air in an almost saturated atmosphere is further cooled radiatively or mixed with colder air of the inversion layer (Fujita and Abe, 2006; Stenni et al., 2016; Walden et al., 2003). Much less frequently, synoptically caused snowfall occurs, which typically has daily amounts an order of magnitude higher than diamond dust precipitation. Thus a few snowfall events per year can yield up to 50% of the annual accumulation. The occurrence of such event-type precipitation is closely related to amplification of Rossby waves, sometimes with corresponding blocking anticyclones (e.g. Massom et al., 2004), and is thus more frequent in the negative state of the SAM or positive ZW3 index (Raphael, 2003; Schlosser et al., 2016), when meridional exchange of heat and moisture is increased. This agrees well with the pattern in Fig. 4a, which shows three distinct maxima in the correlation of SMB and the 850 hPa geopotential height above the Southern Ocean between ca. 60°S and 45°S, south of South Africa, Australia and west of South America, respectively. This spatial pattern is most likely due to the above mentioned positive ZW3 index related to distinct troughs and ridges in the westerlies, which cause advection of relatively warm and moist air to the interior of the continent (Noone et al., 1999; Schlosser et al., 2008; 2010a; 2010b; Massom et al., 2004). There is no straightforward correlation with sea ice extent (Fig. 5a).

Ice and firn cores from EAP do not exhibit a uniform temporal trend. At Kohnen station (EPICA DML drilling site), which is situated at 2892m elevation at the slope, an increase in SMB was found in the past two centuries (Oerter et al., 2000; Altnau et al., 2015), which is parallel to an increase in temperature (derived from stable isotopes), thus most likely caused thermodynamically (Altnau et al., 2015). Anschuetz et al. (2009; 2011) found no clear overall trend in SMB based on data from a traverse from coastal DML to South Pole, with an increase at some sites, and a decrease at others. However, above 3200m, all sites exhibited a decrease in SMB since 1963. Fujita et al. (2011) report that SMB data from a traverse between Dome Fuji and EPICA DML is strongly influenced by topography and is found to be approximately 15% higher in the second half of the 20[th] century than averages over centennial or millennial averages. No significant trend was found in accumulation rates in up to 100yr old cores before 1996 on Amundsenisen in DML (Oerter et al., 1999). The Dome C area (Frezzotti et al., 2005; Urbini et al., 2008) has exhibited high accumulations since the 1960s, as observed along the traverse between Dome Fuji and EPICA DML, whereas no significant changes have been apparent at Dome A since 1260 (Ding et al., 2011a; Hou et al., 2009). In Talos Dome area, at the border between EAP and VL, century-scale variability shows a



slight increase (of a few percent) in accumulation rates over the last 200 years, in particular, since the 1960s, as compared with the period 1816 – 1965 (Frezzotti et al., 2007).

### 3.2.2. Wilkes Land Coast (WL)

5 The snow accumulation regime of Law Dome is determined by the intensity of onshore transport of maritime air masses by cyclonic activity (Bromwich 1988).  While the magnitude of snow accumulation varies along the Wilkes Land Coast, the accumulation is temporally coherent at least as far away as the Shackleton Ice Shelf (Roberts et al, 2015).  In general, this region shows accumulation variability associated with both the El Niño-Southern Oscillation (ENSO) and IPO (Roberts et al., 2015, Vance et al, 2015) which influence the meridional component of the large scale circulation (van Ommen and 10 Morgan, 2010; Roberts et al 2015, Vance et al., 2015).

Law Dome sea ice proxies (Curran et al., 2003; Vallelonga et al., 2017) are correlated with observations of sea ice extent. The weak negative correlation between WL accumulation and sea ice concentration (Fig. 5b) may be indicative of a common forcing from cyclonic systems depositing snow over Law Dome while also contributing to local sea ice break-up and/or 15 dispersion.

### 3.2.3. Weddell Sea Coast (WS)

The WS sector appears to exhibit a positive relationship with sea level pressure (SLP) in the South Pacific, and over the Antarctic continent, while a negative relationship exists over the south Indian Ocean (Fig. 4c). However, this is based on just 20 one ice core which is poorly related to modelled SMB or precipitation. Figure 5c suggests snow accumulation is associated with sea ice concentration in the Weddell Sea, possibly resulting in enhanced moisture availability or reflecting an anticyclone that could draw more northerly air masses to the site. Unfortunately the reduced period of overlap for this site (1980-1992) makes the interpretation less reliable.

### 3.2.4. Antarctic Peninsula (AP)

SMB on the AP exhibits a large west-to-east gradient, exceeding 3000mm w.e. yr$^{-1}$ on the western coast and less than 500 mm w.e. yr$^{-1}$ on the eastern coast [Van Wessem et al., 2016], largely controlled by orography. The AP has been underrepresented in previous SMB studies. However, recent drilling efforts have greatly increased the spatial coverage, but the high annual snowfall still limits the temporal coverage in this region. SMB in AP is dominated by the pattern of SLP in 30 the Amundsen Sea (Fig. 4d), a region of high synoptic activity and the largest contributor of the total Antarctic meridional moisture flux (Tsukernik and Lynch, 2013), with the highest interannual and seasonal variability. Ice cores in this region reveal a significant increase in snowfall during the 20$^{th}$ century (Thomas et al., 2008; 2015, Goodwin et al., 2015), that has





been linked to the positive phase of the SAM and the ASL. High snow accumulation is associated with reduced regional SLP, leading to strengthened circumpolar westerlies and enhanced northerly flow. The mechanism of lower SLP in the Amundsen Sea sector creates a dipole pattern of enhanced precipitation in Ellsworth Land and reduced precipitation over western West Antarctica (Thomas et al., 2015), reflecting the clockwise rotation of air masses and moisture advection paths,

explaining the dipole of correlations observed in Fig. 2 and 3.

Snow accumulation in the Antarctic Peninsula is also closely related to sea ice conditions in the Bellingshausen Sea (Fig. 5d). Sea ice plays an important role in the climate system, acting as a barrier for the transport of moisture and heat between the ocean and the atmosphere. Sea ice reconstructions have revealed a $20^{th}$ century decline in sea ice in the Bellingshausen

Sea (Abram et al., 2010) and evidence that the current rate of sea ice loss is unique for the post-1900 period (Porter et al., 2016). The result is enhanced availability of surface level moisture and increased poleward atmospheric moisture transport (Tsukernik and Lynch, 2013). This was used to explain the longitudinal differences between AP ice core sites (Thomas et al., 2015), with the least significant changes in accumulation in Ellsworth Land and the greatest changes observed in the southwestern AP, where adjacent sea ice exhibits the largest decreasing trend (Turner et al., 2009).

### 3.2.5. West Antarctic Ice Sheet (WAIS)

The WAIS region has been comparatively well-sampled (Fig. 1) with records covering a large portion of the area and spanning many centuries. Although accumulation rates here are relatively high, firn cores recovered as part of the WAIS Divide ice core project have provided records covering the past 2000 years. Because of its complex topography and divide

geometries, the WAIS region is commonly differentiated into two smaller regions: the Amundsen Sea sector, home to Pine Island and Thwaites Glaciers, and the Ross Sea sector, where the Siple Coast ice streams flow into the Ross Ice Shelf.

The low elevation terrain of the Amundsen Sea sector penetrates far into the interior; thus, the region is subject to frequent warm, marine air intrusions that bring cloud cover, higher amounts of snowfall, and higher temperatures (Nicolas and

Bromwich, 2011). In general, the accumulation gradient is dependent on elevation, however, terrain geometry plays an important role as well. The steep coastal region of Marie Byrd Land receives relatively high accumulation because of orographic lifting, while the region directly south, in the precipitation shadow of the Executive Committee Range, is precipitation starved. Thus, the highest accumulation rates are found on the low elevation coastal domes and much of the interior of the Amundsen Sea sector, where marine air intrusions bring moisture and heat (Nicolas and Bromwich, 2011).

Because of the ice sheet geometry, the accumulation records from the Amundsen Sea sector are poorly correlated with those from the Ross Sea sector. Correlation of the WAIS record with RACMO2.4 SMB and ERA-Interim precipitation shows a very strong resemblance to the Amundsen Sea sector and very weak relationship with the Ross Sea sector (Fig. 2e & 3e).



Even though there are records from the Ross Sea sector (Fig. 1), they are largely out of date, covering up to the mid-to-late 1990s. Records from the Amundsen Sea sector provide data up to 2010; thus, the most recent decade of the WAIS record is composed of records only from the Amundsen Sea sector, which has a stronger correlation with modelled SMB. This limitation must be considered when evaluating the drivers of WAIS accumulation.

Higher atmospheric pressure over the Drake Passage brings enhanced northerly flow over the central Amundsen Sea and directs warm, moist air to the low elevation central WAIS (Fig. 4e & 6e). The additional accumulation brought to the Amundsen Sea sector extends deep into the Antarctic interior reaching the South Pole sector (Fig. 2e). This scenario is more representative of the Amundsen Sea rather than the Ross Sea sector, the latter being largely driven by the strength and

position of the Amundsen Sea Low (Kaspari et al., 2004; Nicolas and Bromwich, 2011; Thomas et al., 2015).

### 3.2.6. Victoria Land (VL)

For the purpose of this paper, we cluster records derived from the Ross Ice Shelf and from the coastal regions (below 2,000m elevation) of the Trans-Antarctic Mountains (TAM) into VL (Fig. 1). Despite its geographical diversity, ranging from a low

lying, flat ice shelf to the east (towards West Antarctica) and a steep relief in the west (towards East Antarctica), this region is dominated by cyclonically driven snow accumulation, sensitive to tropical and local climate drivers that significantly impact the wider region (Bertler et al., submitted, Emanuelsson et al., in review).

Accumulation in northern Victoria Land, where the coast faces north, comes primarily from storms in the Indian Ocean

(Scarchilli et al., 2011), the origin of air masses similar to that of adjacent Wilkes Land (Scarchilli et al., 2011). However, the TAM block flow to southern Victoria Land, such that this region is influenced by storms that cross the Ross Sea (Sodeman and Stohl, 2009). As a consequence, the snow accumulation rates are higher in Northern and Southern Victoria Land, while the middle region (including the McMurdo Dry Valleys) lies in precipitation shadows of cyclones from the north and the south, experiencing overall lower snow accumulation rates (Sinclair et al., 2010).

Accumulation across the Ross Ice Shelf and southern VL is linked to that of the AP and WAIS, through the position and intensity of the ASL, which affects the frequency and trajectory of storms in the Ross Sea (Raphael et al., 2016; Turner et al., 2013, Fogt et al., 2012, Bertler et al., 2004). Markle et al., (2012) also found that the phase of ENSO, but not SAM, affected the frequency of Ross Sea storms reaching southern VL. In particular blocking (ridging) events, whose position and

frequency are determined by the intensity and position of the ASL (Renwick et al. 2005), have been identified as a major driver of snow precipitation in the eastern Ross Sea region, enhancing meridional flow across the eastern Ross Sea (Emanuelsson et al., in review). The principal tropical teleconnection associated with the Rossby wave propagation from the western tropical Pacific exerts its influence on the entire Ross Sea region, the Pacific-Southern-American 2 (PSA2) pattern,



which originates from the central, tropical Pacific, shows a distinct, opposing influence between the eastern and western Ross Sea (Raphael et al., 2016, Bertler et al., submitted) leading to a secondary climate signal which explains some of the differences observed in these two regions.

The availability of local moisture from open water in the Ross Sea is also associated with high accumulation events (Sinclair et al., 2013). Furthermore, riming (deposition) might contribute as much as 28 % of all precipitation events, in particular during winter at sites in the vicinity to polynyas, such as RICE (Tuohy et al., 2015). As rime is poorly captured in reanalysis data, the potentially significant contribution of rime at particular sites, might be an important consideration of understanding regional precipitation differences.

The VL composite has the greatest correlation with precipitation in the northward-facing section of VL and the western portion of the Ross Sea. The correlation in the Transantarctic Mountains is low, likely due to topographic complexity. The VL composite is positively correlated with geopotential heights in the eastern Ross Sea (Fig. 4f), illustrating the connection to the ASL, which is more pronounced for Roosevelt Island than for Talos Dome and Hercules Neve. When the ASL is

shifted to the west, weaker storms more frequently penetrate into VL, rather than being focused in central WAIS.

### 3.2.7. Dronning Maud Land (DML)

Precipitation in coastal DML is closely connected to synoptic activity in the circumpolar trough and related frontal systems. Interannual variability of both temperature and precipitation is influenced by SAM, which partly determines the amount of

meridional exchange of heat and moisture. Generally, precipitation decreases from the coast towards the interior, local maxima can occur at the windward side of topographic features, such as ice rises on ice shelves or steep slopes of the escarpment (Rotschky et al., 2007; Schlosser et al., 2008; Vega et al., 2016). Katabatic winds also influence SMB, especially at the grounding line, at the transition of ice shelf and grounded ice, leading to negative SMB values due to wind erosion and increased evaporation (Sinisalo et al., 2013; Schlosser et al., 2014).

The correlation plot of SMB and 850 hPa geopotential height (Fig. 4g), exhibits three distinct maxima above the Southern Ocean, however, compared to EAP they are situated closer to the coast and shifted in longitude by approximately 30°. This also hints at a ZW3 pattern.  There is also a fairly strong positive correlation with sea ice extent in the northwestern Weddell Sea and a rather weak negative correlation with the area between 0° and 30° E (Fig. 5g). A plausible explanation for this

could be that positive sea ice anomalies in the northern part of the Weddell Sea is often related to a comparably strong southwesterly flow that, taking into account Ekman transport, pushes the ice away from the coast, at the same time new ice is formed close to the coast due to cold air advection and re-opened polynyas (e.g. Schlosser et al., 2011). The compensating



northwesterly flow further east could enhance precipitation in DML. However, this is just a qualitative explanation, more research using model and field data would be necessary to prove this.

Ice and firn cores from coastal DML do not show a uniform trend over the past century. However, in recent decades, all but the core from Derwael Ice rise (Philippe et al., 2016) agree in having a negative trend in SMB, whereas temperatures/stable isotope ratios are increasing or constant (Isaksson and Melvold, 2002; Schlosser et al., 2012; Sinisalo et al., 2013; Schlosser et al., 2014; Altnau et al., 2015; Vega et al., 2016). For some cores this negative trend is found for the last 100 years (e.g. Kaczmarska et al., 2004). This suggests that the SMB/precipitation in DML is influenced by the atmospheric flow conditions, i.e. dynamically rather than thermodynamically as at EAP (Altnau et al., 2015).

### 3.3. Regional precipitation variability over the past 200 years

The standardized regional composites have been converted into records of SMB, based on SMB extracted from RACMO2.4. We use a geometric mean regression technique (Smith, 2009) to convert the unit less standardized regional composites into regional SMB (mm w.e yr$^{-1}$). This method allows for error in both the regional composites and the RACMO2.4 data and has been widely used for other ice core proxies, for example regressing sea ice proxies onto satellite derived records of winter sea ice extent (Abram et al., 2010; Thomas and Abram et al., 2016).

The regional SMB records reveal significant variability during the past 200 years (Fig. 6). With the exception of WL (Fig. 6b), all regions exhibit an increasing trend in SMB since 1800 AD, ranging from +1.8 mm w.e. per decade in AP (Fig. 6d) to just +0.15 mm w.e. per decade in EAP and WAIS (Fig. 6a & 6e). In contrast, WL exhibits a decrease in SMB of 0.7 mm w.e per decade since 1800 AD.

WAIS, DML and VL reveal positive trends in SMB since 1800 AD (0.16, 0.69 and 0.84 mm w.e. per decade respectively), but negative trends during the 20$^{th}$ century (-0.57, -0.33 and -0.59 mm w.e. per decade respectively). These changes have low statistical significance (p>0.1), suggesting fluctuations of this magnitude are not unusual. For WAIS there is an increase in SMB during the most recent decade (2000-2010), with SMB 13 mm (2 %) higher than the reference period, while in VL, the most recent decade represents a 4 % reduction (-26 mm) from the reference period. However, the paucity of records and their limited spatial coverage since the 2000s limits our ability to assess how regionally representative this trend is.

The 20$^{th}$ century increases in AP (Fig. 6d) do appear unusual in the context of the past 200 years. In this region, the running decadal mean during the early 2000s exceeds two standard deviations above the record average for the entire 200-year period. SMB in the AP has been increasing at a rate of 6.6 mm w.e per decade since 1900 (p<0.01), equivalent to a 76 mm (~19 %) increase between the decadal average at the start of the 20$^{th}$ century (1901-1910) and the start of the 21$^{st}$ century



(2001-2010). As observed in previous studies the increase in SMB in the AP began in the ~1930 and accelerated during the 1990s (Thomas et al., 2008; 2015; Goodwin et al., 2015).

The second highest increase is observed in the WS region (Fig. 6c), where SMB has increased at a rate of 1 mm w.e. per decade (p<0.05) since 1800 AD and 2.6 mm w.e. per decade (p<0.05) since 1900 AD (1900-1992). The spatial correlation between the AP composite, with both RACMO2.4 and ERA-interim (Fig 2d & 3d), revealed that the region of significant correlations extends into the WS, suggesting the increases observed in AP may extend over the WS. However, there is some doubt about the WS record capturing regional precipitation and more records are needed to confirm this connection.

In DML, two of the three records which cover the entire 20[th] century reveal a decreasing trend, while the Derwael Ice Rise record reveals a statistically significant increase in snow accumulation during the 20[th] century. Snow accumulation at this low elevation coastal site has been related to sea ice and atmospheric circulation patterns (Philippe et al., 2016), with SMB during the most recent decade (2000-2010) 5 % higher than the reference period.

### 3.4. Regional precipitation variability over the past 1000 years

To assess the significance of the recent trends we extend the records back 1000 years (Fig. 7). Only the Law Dome, RICE, WAIS and Berkner ice cores cover the full period and the increased variability in these regions beyond ~1400 AD is an artefact of the shift from multiple to single sites.

There is considerable interannual and multi-decadal variability in all records, however, there is little consistency or commonality between regions. Previous studies have related Antarctic SMB changes to solar irradiance (Frezzotti et al., 2013), with three periods of low accumulation variability identified at 1250-1300, 1420-1550 and 1660-1790 AD that correspond to periods of low solar activity. A decrease in snow accumulation for the entire Little Ice Age Period (defined as ~1300-1800 AD) was also observed in the western Ross Sea by a lower resolution record (Bertler et al., 2011). Examination of our regional composites over the past 1000 years indicates that this may be restricted to the EAP and perhaps small regions not captured in this array, with little evidence for a large scale reduction in variability related to solar variability in other regions. The relatively low accumulation over EAP, combined with the major (>50 %) contribution of non-synoptic precipitation, may explain the stronger influence of solar variability over EAP while local wind regimes in the TAM have been shown to be sensitive to solar radiation (Bertler et al., 2005). However, the updated network presented here suggests that the strong synoptic influence on coastal regions is likely to outweigh any direct solar influence.

Histograms of running 50-year and 100-year trends are shown in Figure 8, with the last 50 and 100-year trends of the 20[th] century highlighted. With the exception of AP (Fig. 8d), the trends during the late 20[th] century are within the expected range

of variability in the context of the past 1000 years (300 yrs for AP and DML, 770 yrs for EAP). However, in AP the 50 and 100-year trends at the end of the 20th century appear unusual. Both the highest 50 and 100-year trends occur during the most recent period (1960-2010 and 1910-2010), suggesting the rate of increase in snowfall during the late 20th and early 21st century in this region is exceptional in the context of the past 300 years. However, the shorter records in AP (and DML) reduce the significance of the recent trends here relative to the other regions.

## 4.  Total Antarctic SMB change

An estimate of the SMB change for the grounded AIS is calculated by weighting the regional SMB composites relative to their total area (as defined in Fig. 1). The area weighted SMB change is calculated as the sum of the "area-weighted" regional SMB composites, divided by the area of Antarctica (~12 million km$^2$).

The total Antarctic SMB has increased by ~ 44 GT w.e. between 1800 AD and 2000 AD. The largest (area weighted) increase is observed in the AP, accounting for ~72 % of the total Antarctic SMB increase. Despite the relatively low annual snowfall, the EAP is the second highest contributor (22 %), due to its large area (accounting for 45 % of the total area).

Only four records cover the full 1000 years (Fig. 7). SMB in these regions (WAIS, WL, WS and VL) decreased by 39 GT w.e. since 1000 AD. However, it is important to note that these four regions represent just ~30 % of the total area of Antarctica and in the case of WS, doubt exists as to how regionally representative the record is.  Evidence from the 20th century suggests that even small changes in SMB, in either the low accumulation/ high area EAP region or the high accumulation AP region, can have significant impacts on the total Antarctic SMB budget.

## 5.  Conclusions

As part of the Antarctica 2k community effort we present regional snow accumulation composites to investigate snow accumulation variability over the past 1000 years. Eighty ice core snow accumulation records were, quality checked and separated into seven geographical regions, to reduce the bias towards over-represented regions and separate the high accumulation coastal zones from the low accumulation high elevation sites.

The snow accumulation records from each region were evaluated against SMB from RACMO2.4 and precipitation from ERA-interim. With the exception of the EAP and WS region, the regional composites capture a large proportion of the regional SMB and precipitation variability. The lack of correlation in the EAP is likely due to the greater influence of wind

(erosion or deposition), sublimation, and post-depositional processes (surface glazing and dune formation) in the interior than in coastal regions. Another explanation is that the models cannot capture processes such as deposition of diamond dust, likely having a relatively large influence on precipitation variability in these extremely cold areas (van de Berg et al. 2005, Mahesh at al. 2001). Either way, the lack of coherency in trends from ice core records across the central Antarctic Plateau

suggests that ice core snow accumulation records from this region are less well suited to studies investigating temporal changes in Antarctic SMB.

Our study suggests an overall increase in SMB across the grounded Antarctic ice sheet since 1800 AD, of approximately 44 GT, with the largest (area-weighted) contribution from the AP. This contradicts some previous studies which suggest a

negligible change in Antarctic SMB since 1957 (Monaghan et al., 2006) and that the current SMB is not exceptionally high compared to the past 800 years (Frezzotti et al., 2013). The later study did acknowledge high SMB in coastal regions since the 1960s, however, these studies were hindered by a lack of recent records (especially from the coastal zones) and were heavily weighted by the EAP region.  Recent drilling campaigns, and the collation of records as part of the Antarctica 2k program, have allowed us to better represent several important regions and derive a more accurate representation of SMB

changes over the past 200 years.

The inclusion of new snow accumulation records that cover the past 1000 years have provided valuable information about SMB changes in certain regions, however estimating Antarctic SMB using these sites alone would be misleading. For example, based on the four available records which cover the past 1000 years there is evidence for a decreasing trend in

SMB since 1000 AD. However, the combined regional representation of these records is less than 30 % of the total Antarctic continent and includes single ice core sites with only limited regional representation in SMB. Our findings suggest that small changes in the high accumulation AP, or the low accumulation but geographically much larger EAP region, could change the sign and significance of the total Antarctic SMB trend dramatically.

Greater spatial representation (especially WS and EAP) and the inclusion of sufficiently deep ice cores from high-accumulation coastal zones (especially AP and DML) are vital to our understanding the true nature of Antarctic SMB in the past and providing an accurate benchmark for how SMB may change in the future.





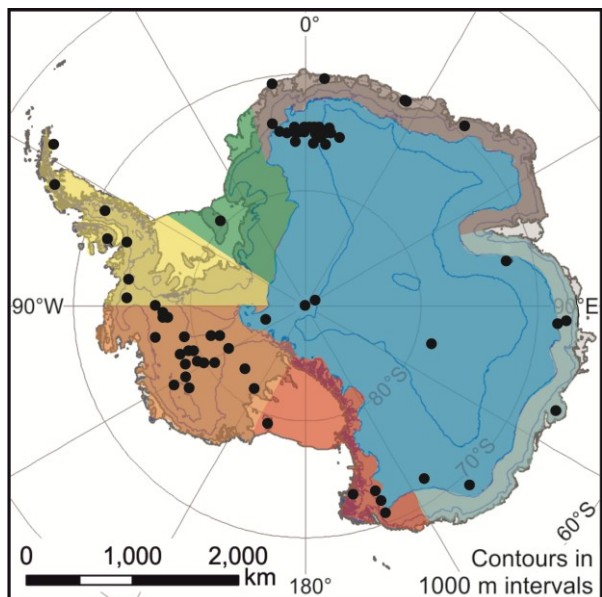

Figure 1: Location of all ice cores and the regional boundaries used in this study. EAP (blue), WL (cyan), WS (green), AP (yellow), WAIS (orange), VL (red) and DML (brown).



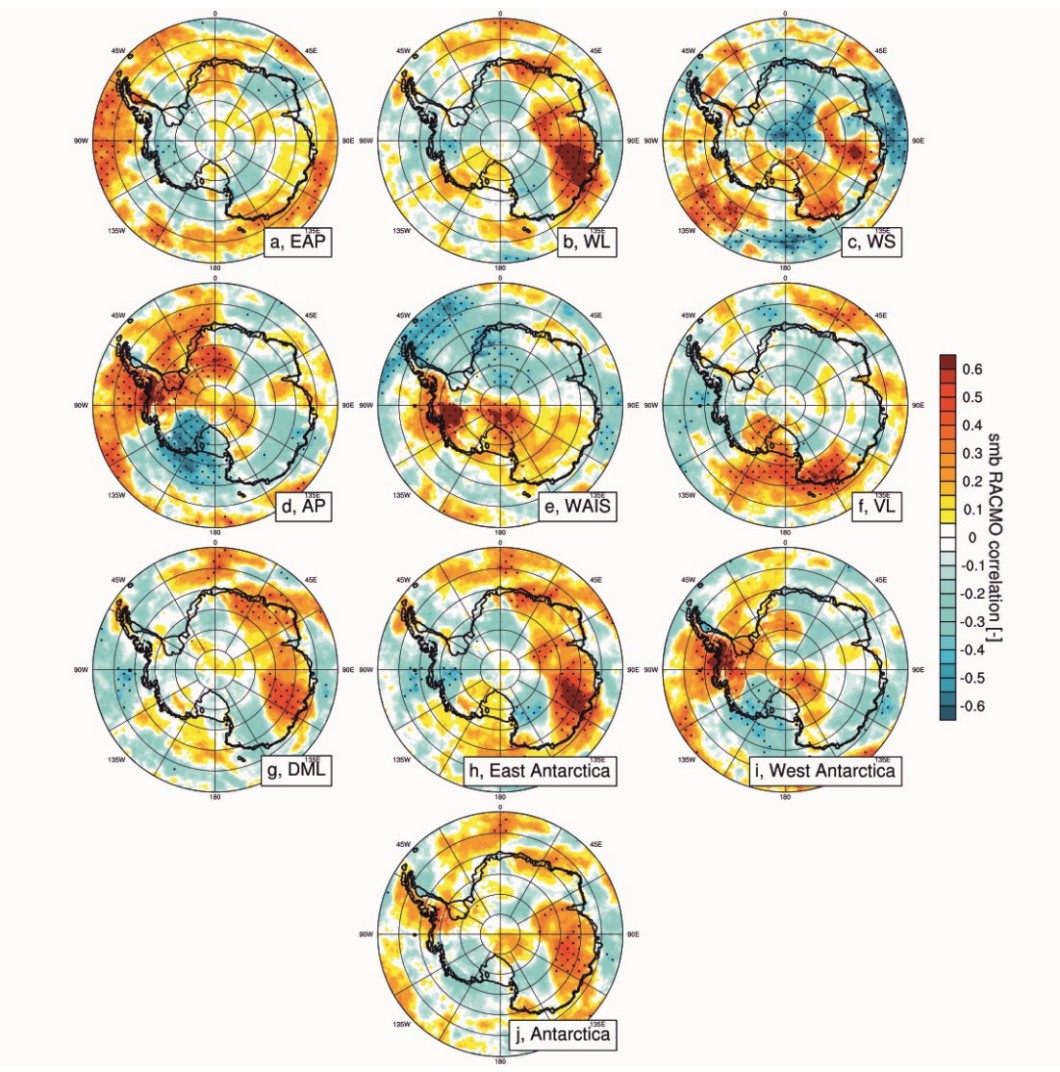

**Figure 2: Spatial correlation plots of standardized regional and continental composites of snow accumulation with SMB from RACMO2.4 (1979-2010). Grid points with >95 % significance are dotted. Note, correlation with WS only cover the period 1979-1992.**





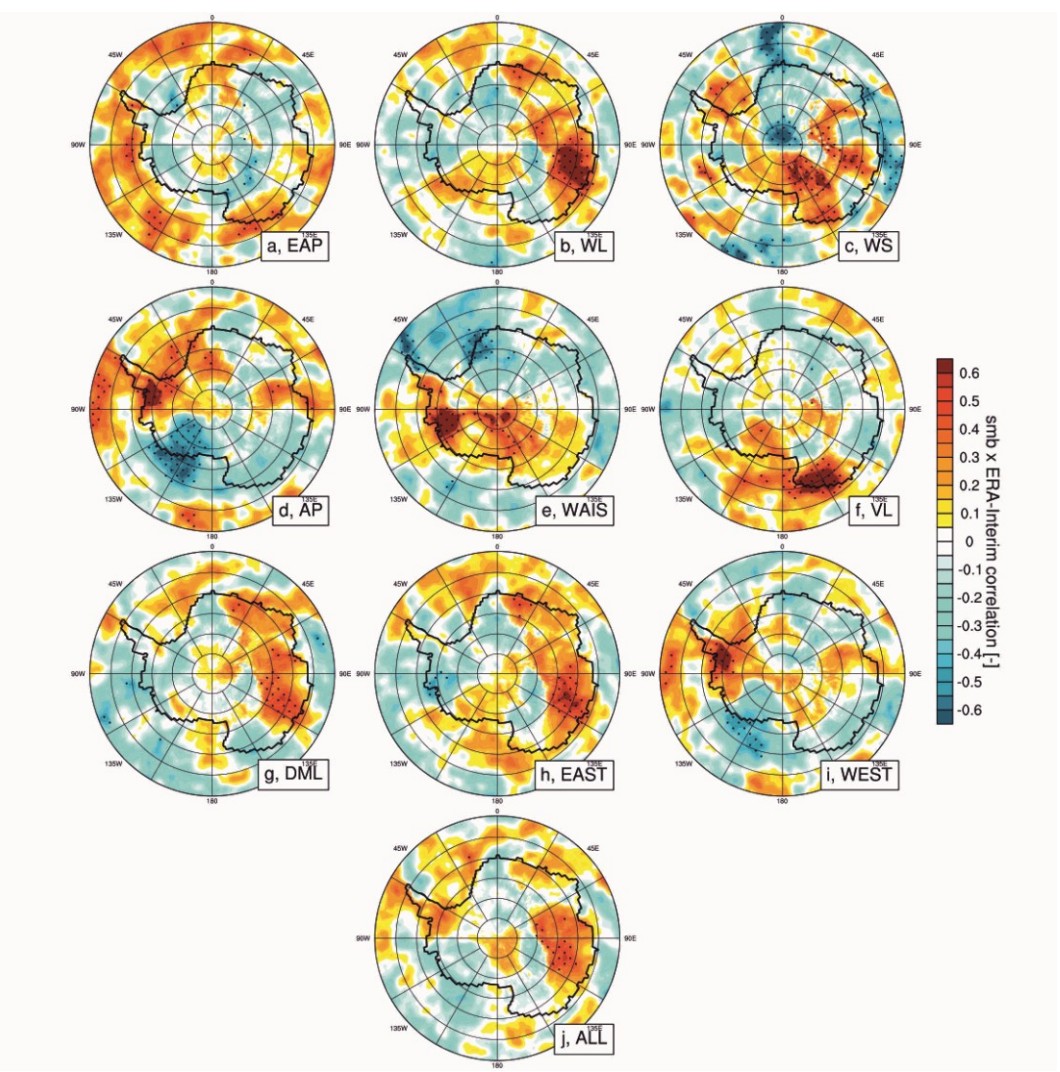

**Figure 3: Spatial correlation plots of standardized regional and continental composites of snow accumulation with precipitation from ERA-interim (1979-2010). Grid points with >95 % significance are dotted. Note, correlation with WS only cover the period 1979-1992.**





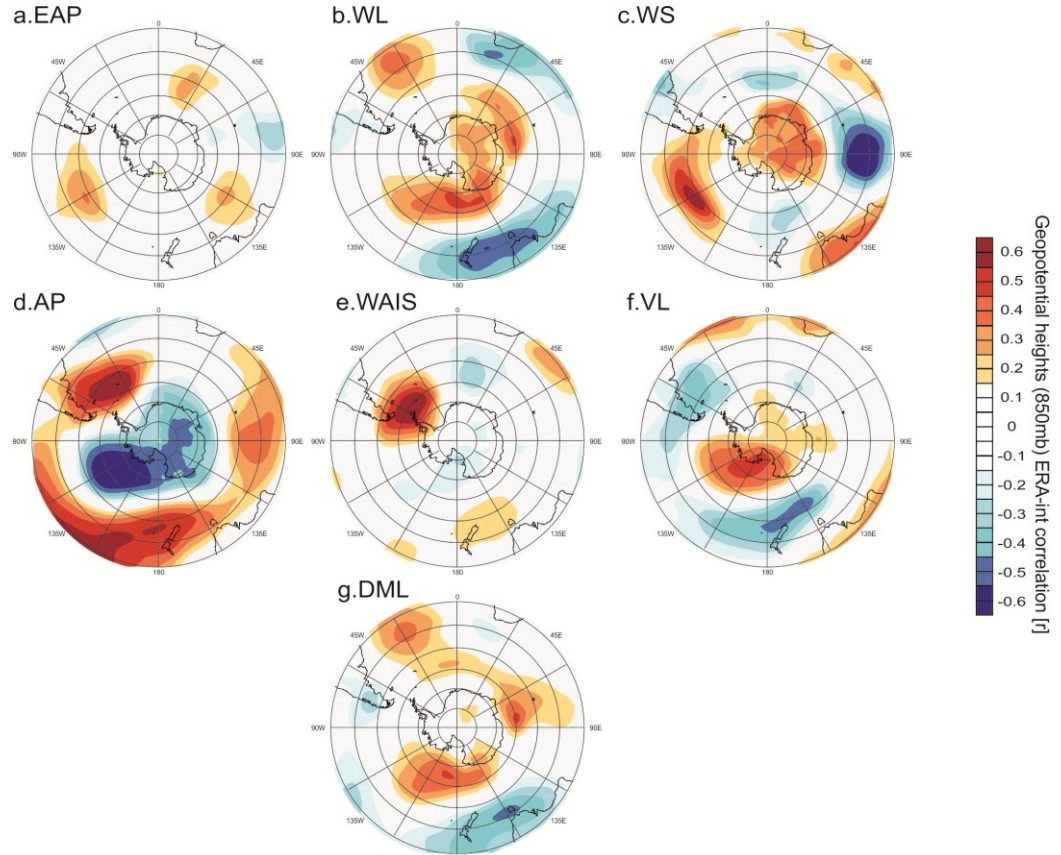

**Figure 4: Spatial correlation plots of standardized regional snow accumulation composites with annual 850 hPa geopotential heights from ERA-interim (1979-2010). Note, correlation with WS only cover the period 1979-1992.**




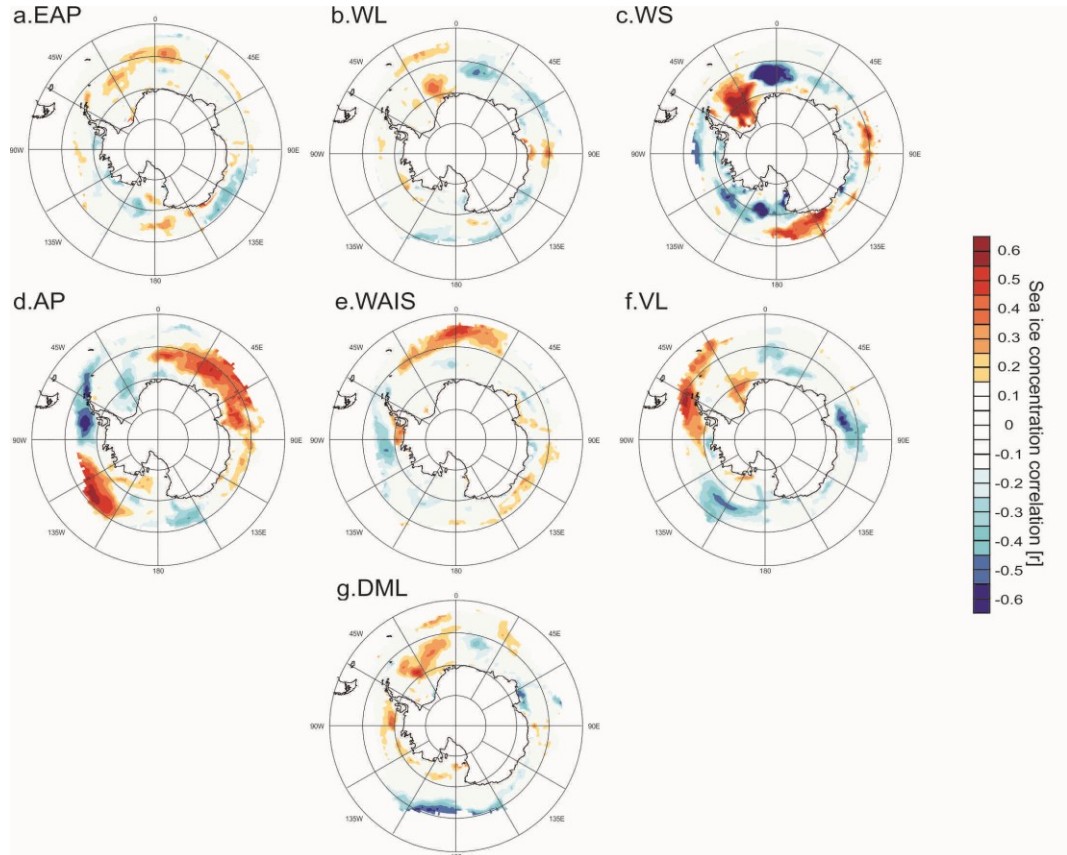

**Figure 5: Spatial correlation plots of standardized regional snow accumulation composites with annual sea ice concentration from bootstrap analysis (1981-2010). Note, correlation with WS only cover the period 1979-1992.**





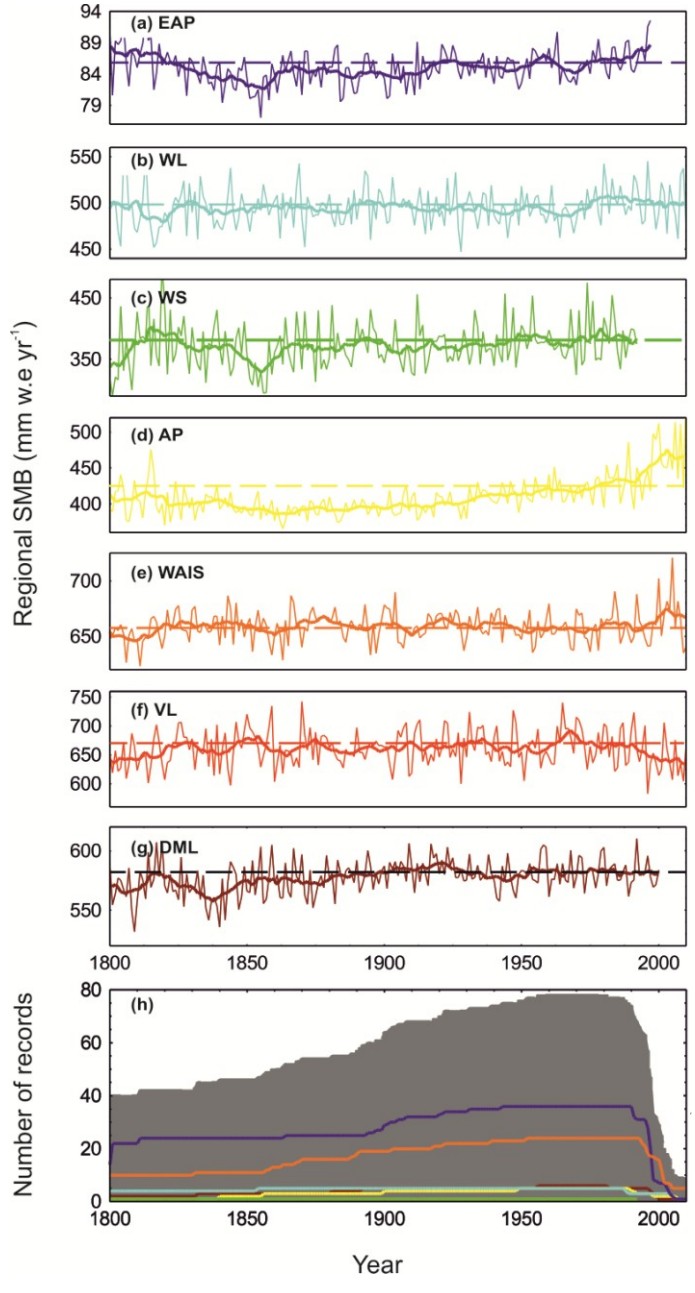

Figure 6: Regional SMB composites (1800-2010 AD) shown as annual averages (thin lines) and 5 year means (thick lines). Plots 1-7 East Antarctic Plateau (EAP dark blue), Wilkes Land Coast (WL cyan), Weddell Sea Coast (WS green), Antarctic Peninsula (AP yellow), West Antarctic Ice Sheet (WAIS orange), Victoria Land (VL red) and Dronning Maud Land (DML brown). Bottom plot represents the total number of records (solid grey) and the number of records by region.




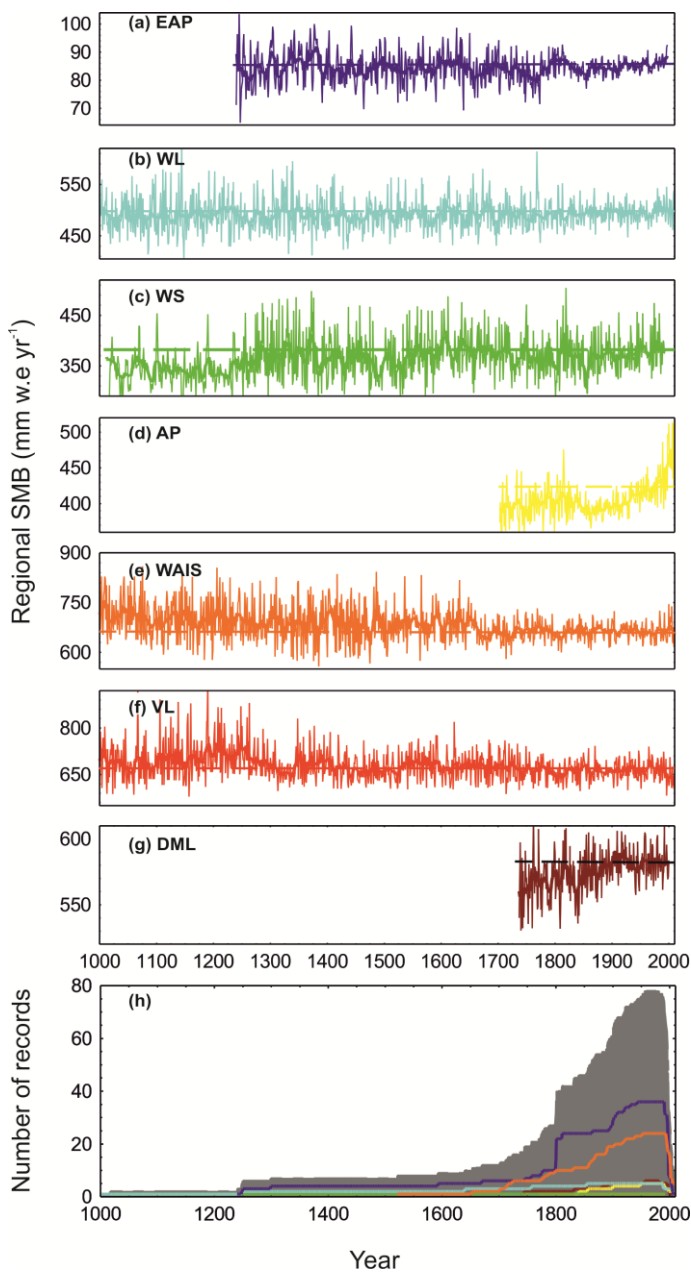

Figure 7: Regional SMB composites (1000-2010 AD) shown as annual averages (thin lines) and 10 year running means (thick lines). Plots 1-7 East Antarctic Plateau (EAP dark blue), Wilkes Land Coast (WL cyan), Weddell Sea Coast (WS green), Antarctic Peninsula (AP yellow), West Antarctic Ice Sheet (WAIS orange), Victoria Land (VL red) and Dronning Maud Land (DML brown). Bottom plot represents the total number of records (solid grey) and the number of records by region.





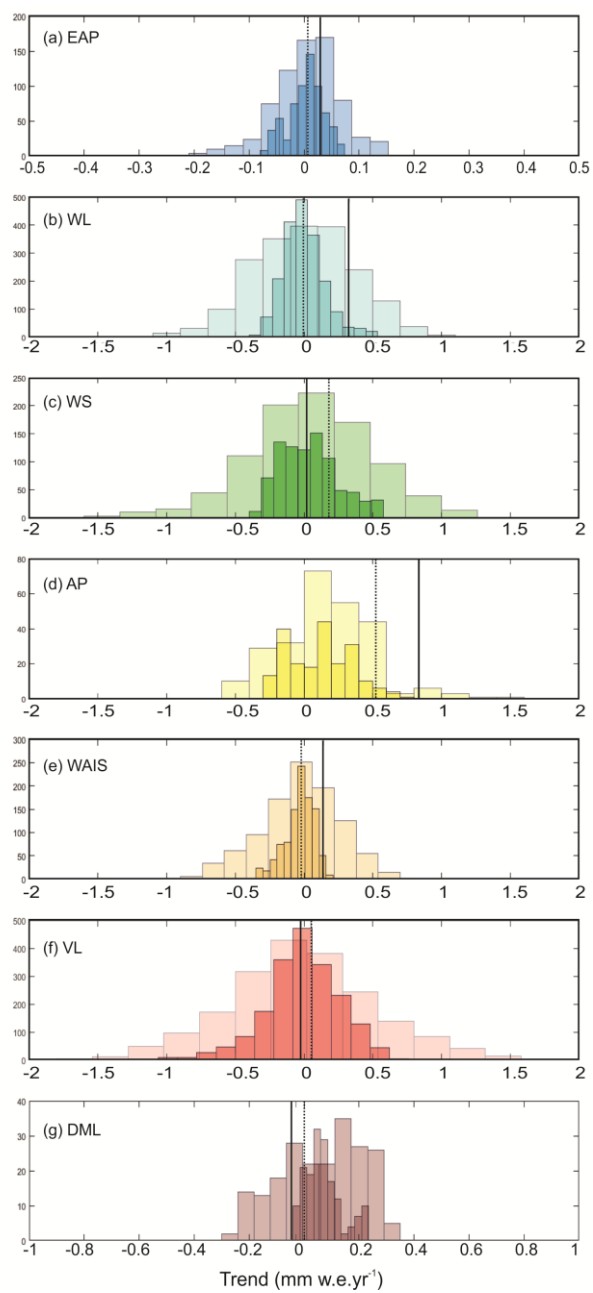

**Figure 8: Histogram of running 50-year (lighter shading) and 100-year (darker shading) trends for each region (mm w.e. yr$^{-1}$).**
**Solid vertical line represents the most recent 50-year trend and dashed vertical line the most recent 100-year trend (respectively,**
**2010-1961 and 2010-1911).**



6.  **Data availability:** All data presented in this study is available at the Antarctica 2k database (http://www.pages-igbp.org/ini/wg/antarctica2k/data). The composite records are also available from the UK Polar Data Centre (www.bas.ac.uk/data/uk-pdc) or by contacting Elizabeth Thomas (lith@bas.ac.uk).

7.  **Acknowledgments:** This work was supported by the PAGES Antarctica 2k working group and funded by the British Antarctic Survey (Natural Environment Research Council). We would like to thank the Antarctic ice core community for making their snow accumulation data available to the Antarctica 2k database and thank the data managers and PAGES team for their support.

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





| Site Name | Latitude | Longitude | Elevation (m) | Years AD | Reference |
|---|---|---|---|---|---|
| Vostok composite VRS13 | -78.47 | 106.83 | 3488 | 1654-2010 | Ekaykin et al., 2014 |
| B31Site DML07 | -75.58 | -3.43 | 2680 | 1000-1994 | Graf et al., 2000 |
| B32Site DML05 | -75 | -0.01 | 2892 | 1248-1996 | Graf et al., 2000 |
| B33Site DML17 | -75.17 | 6.5 | 3160 | 1250-1997 | Graf et al., 2000 |
| FB96DML01 | -74.86 | -2.55 | 2817 | 1895-1995 | Oerter et al., 1999 |
| FB96DML02 | -74.97 | -3.92 | 3014 | 1919-1995 | Oerter et al., 1999 |
| FB96DML03 | -74.49 | 1.96 | 2843 | 1941-1996 | Oerter et al., 1999 |
| FB96DML04 | -74.40 | 7.21 | 3161 | 1905-1996 | Oerter et al., 1999 |
| FB96DML05 | -75.00 | 0.012 | 2882 | 1930-1996 | Oerter et al., 1999 |
| FB96DML06 | -75.01 | 8.01 | 3246 | 1899-1996 | Oerter et al., 1999 |
| FB96DML07 | -74.59 | -3.439 | 2669 | 1908-1996 | Oerter et al., 1999 |
| FB96DML08 | -75.75 | 3.2936 | 2962 | 1919-1996 | Oerter et al., 1999 |
| FB96DML09 | -75.93 | 7.2217 | 3145 | 1897-1996 | Oerter et al., 1999 |
| FB96DML10 | -75.22 | 11.35 | 3349 | 1900-1996 | Oerter et al., 1999 |
| FB9803 | -74.85 | -8.497 | 2600 | 1921-1997 | Oerter et al., 1999 |
| FB9804 | -75.25 | -6.0 | 2630 | 1801-1996 | Oerter et al., 1999 |
| FB9805 | -75.16 | -0.99 | 2840 | 1800-1997 | Oerter et al., 1999 |
| FB9807 | -74.99 | 0.036 | 2880 | 1758-1997 | Oerter et al., 1999 |
| FB9808 | -74.75 | 0.999 | 2860 | 1801-1997 | Oerter et al., 1999 |
| FB9809 | -74.49 | 1.960 | 2843 | 1801-1997 | Oerter et al., 1999 |
| FB9810 | -74.66 | 4.001 | 2980 | 1801-1997 | Oerter et al., 1999 |
| FB9811 | -75.08 | 6.5 | 3160 | 1801-1997 | Oerter et al., 1999 |
| FB9812 | -75.25 | 6.502 | 3160 | 1810-1997 | Oerter et al., 1999 |
| FB9813 | -75.17 | 5.00 | 3100 | 1800-1997 | Oerter et al., 1999 |
| FB9814 | -75.08 | 2.50 | 2970 | 1801-1997 | Oerter et al., 1999 |
| FB9815 | -74.96 | -1.50 | 2840 | 1801-1997 | Oerter et al., 1999 |
| FB9816 | -75 | -4.51 | 2740 | 1800-1997 | Oerter et al., 1999 |
| FB9817 | -75.00 | -6.49 | 2680 | 1800-1997 | Oerter et al., 1999 |
| South Pole 1995 | -90 | 0 | 2850 | 1801-1991 | Mosely-Thomposn et al., 1999 |
| GV2 | -71.71 | 145.26 | 2143 | 1670-2003 | Frezzotti et al., 2004; 2013 |
| D66 | -68.94 | 136.94 | 2333 | 1864-2003 | Frezzotti et al., 2004; 2013 |
| LGB65 | -71.85 | 77.92 | 1850 | 1745-1996 | Xiao et al., 2004 |



| | | | | | |
|---|---|---|---|---|---|
| US-ITASE-2002-7 | -88.99 | 59.97 | 3000 | 1900-2002 | Mayewski and Dixon, 2013 |
| US-ITASE-2002-4 | -86.5 | -107.99 | 2586 | 1594-2003 | Mayewski and Dixon, 2013 |
| DSS Law Dome | -66.77 | 112.81 | 1370 | 0-1995 | Roberts et al., 2015 |
| 105th km | -67.43 | 93.38 | 1407 | 1757-1987 | Vladimirova et al., 2014 |
| 200th km | -68.25 | 94.08 | 1990 | 1640-1988 | Ekaykin et al., 2016 |
| Berkner Island (South) | -79.57 | -45.72 | 890 | 1000-1992 | Mulvaney et al., 2002 |
| Gomez | -73.59 | -70.36 | 1400 | 1858-2006 | Thomas et al., 2008 |
| James Ross Island | -64.22 | −57.68 | 1640 | 1832-1997 | Aristarain et al., 2004 |
| Dyer Plateau | -70.68 | -64.87 | 2002 | 1505-1988 | Thompson et al., 2004 |
| Bruce Plateau | -66.04 | -64.08 | 1976 | 1900-2009 | Goodwin et al., 2016 |
| Beethoven | -71.9 | 74.6 | 580 | 1949-1991 | Pasteur and Mulvaney, 2000 |
| Ferrigno | -74.57 | -86.9 | 1354 | 1703-2010 | Thomas et al., 2015 |
| Bryan Coast | -74.50 | -81.68 | 1177 | 1712-2010 | Thomas et al., 2015 |
| DIV2010 | -76.77 | -101.74 | 1330 | 1786-2010 | Medley et al., 2013 |
| THW2010 | -76.95 | -121.22 | 2020 | 1867-2010 | Medley et al., 2013 |
| PIG2010 | -77.96 | -95.96 | 1590 | 1918-2010 | Medley et al., 2013 |
| WDC05A | -79.46 | -112.09 | 1806 | 1775-2005 | Banta et al., 2008 |
| WD05Q | -79.46 | -112.09 | 1759 | 1522-2005 | Banta et al., 2008 |
| WAIS 2014 | -79.46 | -112.09 | 1759 | 0-2006 | Fudge et al., 2016 |
| CWA-A | -82.36 | -119.286 | 950 | 1939-1993 | Reusch et al., 1999 |
| CWA-D | -81.37 | -107.275 | 1930 | 1952-1993 | Reusch et al., 1999 |
| Siple dome-94 | -81.65 | -148.79 | 620 | 1891-1994 | Kaspari et al., 2004 |
| Upstream-C (UP-C) | -82.44 | -135.97 | 525 | 1870-1996 | Kaspari et al., 2004 |
| Ross ice drainage system A | -78.73 | -116.33 | 1740 | 1831-1995 | Kaspari et al., 2004 |
| Ross ice drainage system B | -79.46 | -118.05 | 1603 | 1922-1995 | Kaspari et al., 2004 |
| Ross ice drainage system C | -80.01 | -119.43 | 1530 | 1903-1995 | Kaspari et al., 2004 |
| US-ITASE-1999-1 | -80.62 | -122.63 | 1350 | 1724-2000 | Kaspari et al., 2004 |
| US-ITASE-2000-1 | -79.38 | -111.24 | 1791 | 1673-2001 | Kaspari et al., 2004 |
| US-ITASE-2000-3 | -78.43 | -111.92 | 1742 | 1971-2001 | Mayewski and Dixon, 2013 |
| US-ITASE-2000-4 | -78.08 | -120.08 | 1697 | 1798-2000 | Kaspari et al., 2004 |
| US-ITASE-2000-5 | -77.68 | -123.99 | 1828 | 1718-1999 | Kaspari et al., 2004 |
| US-ITASE-2001-1 | -79.16 | -104.97 | 1842 | 1986-2002 | Mayewski and Dixon, 2013 |
| US-ITASE-2001-2 | -82.00 | -110.01 | 1746 | 1892-2002 | Kaspari et al., 2004 |



| US-ITASE-2001-3 | -78.12 | -95.65 | 1620 | 1858-2002 | Kaspari et al., 2004 |
|---|---|---|---|---|---|
| US-ITASE-2001-4 | -77.61 | -92.24 | 1483 | 1986-2001 | Mayewski and Dixon, 2013 |
| US-ITASE-2001-5 | -77.06 | -89.14 | 1239 | 1780-2002 | Kaspari et al., 2004 |
| Hercules Névé | -73.1 | 165.4 | 2960 | 1770-1992 | Stenni et al., 2000 |
| TD96 Talos Dome | -72.8 | 159.06 | 2316 | 1232-1995 | Graf et al., 2002 |
| GV7 | -70.68 | 158.86 | 1947 | 1854-2004 | Frezzotti et al., 2004; 2014 |
| GV5 | -71.89 | 158.54 | 2184 | 1777-2004 | Frezzotti et al., 2004; 2007 |
| RICE | -79.36 | 161.64 | 560 | 0-2012 | Bertler et al., submitted |
| Fimbulisen S20 | -70.25 | 4.82 | 63 | 1956-1996 | Isaksson et al., 1999 |
| Fimbulisen S100 | -70.24 | 4.8 | 48 | 1737-1999 | Kaczmarska et al., 2004 |
| Georg-von-Neumayer (B04) | -70.62 | -8.37 | 28 | 1892-1981 | Schlosser et al., 1999 |
| Kottas Camp FB9802 | -74.21 | -9.75 | 1439 | 1881-1997 | Oerter et al., 2000 |
| H72 | -69.2 | 41.08 | 1214 | 1832-1999 | Nishio et al., 2002 |
| Derwael Ice Rise IC12 | -70.25 | 26.34 | 450 | 1744-2011 | Philippe et al., 2016 |

**Table 1: Ice core data used in this study.**