# Peer review of "Regional Antarctic snow accumulation over the past 1000 years"

_Climate of the Past, 2017_

## Referee Comment (RC1) · Anonymous Referee #1 · 3 May 2017

**Overall comments**

The authors compile some 80 Antarctic ice core records that meet their requirements for temporal coverage, time resolution, and corrections for layer thinning. The records are grouped into regions, composited, and then the regional trends and variability over the past 200-1000 years are discussed. Finally, estimate an overall increase in SMB of ∼44 GT since 1800 AD, with much of it occurring within the past couple decades. In general, the paper is very well written and logically organized. It is hard to find a major fault with this paper. It is an accomplishment just to compile the records, requiring the cooperation of scientists from many nations and reflecting many years of field work. If anything, the paper is a bit too guarded and tentative at times: "However, this is just a qualitative explanation, more research using model and field data would be needed to prove this." or "The reduced period of overlap...makes this interpretation less

reliable." and many other examples. Caveats are of course a natural part of science, but the inclusion of so many in this paper prevents it from being the final word on snow accumulation or even a paper that would get cited a lot (perhaps they are planning a Nature paper that will pack more punch.).

Specific comments

Affiliations, page 1: I think some of the affiliations are incorrect, please check. For instance, I believe B. Medley is at #9 (NASA), not #10 (U Victoria).

Abstract, line 14: increase in SMB across grounded AIS of ∼44 GT since 1800: Some context for this number would be helpful. Is that a lot in terms of mitigating SLR? What is the SLR equivalent? Does this number make sense in terms of published global sea level budgets over the past 100-200 years (is it in the noise or a significant number?)?

Page 4, first paragraph: Given the projected increase in SMB, is it expected to offset overall ice sheet mass loss; is Antarctica expected to be a net contributor to SLR given the overall mass budget?

Page 4, line 30: PAGES Antarctica 2K community: Only a select few readers may know what PAGES is, let alone the 2K community. Define?

Page 6, lines 11-14: Was their any requirement for proven dating precision and accuracy? Are we assuming that all of the records are perfectly dated?

Page 8, lines 19-20: "predominantly positive phase of the IPO/PDO." There was a major shift in the PDO/IPO in 1998-1990, from positive to negative, affecting more than a third of the 1979-2010 period. This has been shown to impact a number of Antarctic climate trends and it may even be reflected in the recent increase in accumulation in the AP and the decrease in VL. So, I don't think it is accurate to say the the IPO/PDO was predominantly in its positive phase.

Pages 13-14: "The principal teleconnection associated with the Rossby wave propagation from the western tropical Pacific...which originates from the central, tropical

[Figure]

Pacific." This sentence is repetitive, as well as contradictory (western Pacific vs central Pacific). A rewrite is needed.

Also, in the discussion of the VL and AP composites (sections 3.24 and 3.26), I can't help but notice that the teleconnection patterns of these two regions are roughly opposite in sign. See Figure 4d and 4f. I'm surprised this isn't mentioned somewhere in the paper. 4d resembles the trend pattern associated with the negative PDO, which could have played a role in the recent increase in AP accumulation and decrease in VL precipitation. I also wonder why tropical teleconnections aren't mentioned with respect to AP accumulation.

Page 15, line 14: change "unit less" to "unitless."

Page 15, bottom two paragraphs: As mentioned above, it would be interesting to discuss the opposing accumulation trends in terms of the PDO phase and/or the ASL deepening trend.

Page 17, line 27: Change "were, quality" to "were quality"

Figures 4 and 5: In contrast to figures 2 and 3, no significance levels are indicated on Figures 4 and 5. Could stippling for significance be added to Figures 4 and 5?

Figure 5 caption, page 23: should be "correlations...cover" or "correlation...covers."

---

## Short Comment (SC1) · 22 May 2017

The PAGES Data Stewardship Integrative Activity seeks to advance best practices for sharing data generated and assembled as part of all PAGES-related activities. As part of this activity, a team of reviewers has been constituted for the "Climate of the Past 2000 years" Special Issue. The data team is reviewing the data handling within each of the CP-Discussion papers in relation to the CP data policy and current best practices. The team has identified essential and recommended additions for each paper, with the goal of achieving a high and consistent level of data stewardship across the 2k Special Issue. We recognize that an additional effort will likely be required to meet the high level of data stewardship envisaged, and we appreciate the dedication and contribution of the authors. This includes the use of Data Citations (see example in supplement). We ask authors to respond to our comments as part of the regular

open interactive discussion. If you have any questions about PAGES Data Stewardship principles, please contact any of us directly.

Best wishes for the success of your paper,

2k Special Issue Data Review Team (Darrell Kaufman, Nerilie Abram, Belen Martrat, Raphael Neukom, Scott St. George) and ex-officio team members (Marie-France Loutre, Lucien von Gunten)

Essential additions for this paper:

(1) Update the "Data Availability" section to include an explicit URL address to locate the input datasets and the reconstructions generated in this study. For the input datasets, we suggest creating a landing page for this study that lists the 80 individual datasets.

(2) Add Data Citations (in addition to publication citations) for each of the 80 datasets used in this study. For those data not already in a persistent public repository, submit essential metadata (including the name of the Antarctica region to which it was assigned in this study) along with the accumulation time series itself and add the Data Citation/URL in Table 1.

(3) If any of the cores used in this study were also used in previous PAGES 2k databases (temperature or isotopes), please include cross references to those IDs.

(4) Submit the primary outcome of this study, the composite accumulation time series by region (Fig 7), to a public repository and include the Data Citation/URL in "Data Availability".

Recommended elements are:

(5) Archive the gridded spatial correlations of snow accumulation data shown in Figs 2 though 5.

(6) Archive the 50- and 100-year trends for each record as shown in Fig 8.

Please also note the supplement to this comment:
http://www.clim-past-discuss.net/cp-2017-18/cp-2017-18-SC1-supplement.pdf

---

## Referee Comment (RC2) · Anonymous Referee #2 · 14 Jul 2017

Thomas et al. have compiled available high-resolution ice core accumulation records from Antarctica, and review those within the PAGES 2k framework. They divide the cores into 7 regions to provide a regional perspective on ice accumulation and surface mass balance in Antarctica. The surface mass balance of Antarctica is an important topic of study, with implications for the Antarctic contribution to sea-level rise. Overall the study is relevant, interesting, well-written and clear. I propose some minor corrections and additions for readability, procedural clarity and a more thorough analysis.

General comments:

1) My first concern is procedural clarity and treatment of uncertainties.

1a) Combining individual records must be done carefully to avoid jumps in the com-

posite at the locations where the number of cores changes. There is some discussion in section 2.6, but it could be expanded. How did you normalize the records? Did you scale the variance, or just the mean? Did all records include the full 1960-1990 reference period?

1b) How are age uncertainties incorporated? Are all chronologies based on annual layer counts? Section 2.3 merely states they need to have annual resolution, so I assume this is the case. The age uncertainties are of course critical when comparing to RACMO and ERA-interim. What is the typical uncertainty in the age scales, and can this influence the conclusions?

Also, dating uncertainties will introduce accumulation uncertainties, because accumulation is essentially the derivative of the depth-age relationship. So a 5% age uncertainty results in a 5% accumulation uncertainty.

1c) I assume that all RACMO and ERA data are annual means? Were these taken as calendar years? The annual proxies used in the layer count could of course represent another time period (e.g. spring to spring).

1d) How was the layer thinning correction done? Was the treatment identical for all records? Section 2.2 reviews several methods (Nye, Dansgaard-Johnsen and Roberts), but it's not stated which one is used. Was this done on a case-by-case basis?

This does not matter for the inter-annual variability, but this is critical for investigating the long-term trends. Details are needed for the reader to evaluate how reliable the trends are. Please elaborate on this.

1e) What is the typical uncertainty in the thinning correction, and how does it influence the reconstructed centennial-scale trends?

2) My second suggestions focus on the analysis of the records

2a) The authors test how representative the records are by comparing them to RACMO

and ERA-interim using spatial correlation plots, which is very qualitative and includes some hand-waving. Since RACMO and ERA-interim are gridded products, it should be trivial to extract the SMB time series directly at the core locations – these model time series could then be composited for the exact same time periods and in the same way as the ice core records were composited. This will allow for easy quantitative analysis. For example, how well are the model and data composites correlated? How well does the model capture variability within each region? Etc. This could complement the figures provided.

2b) How well do individual cores in a region represent the regional composite? It would be very easy to figure out in a principal component analysis. The variance explained by the first component will tell you how much of the signal variance is shared between the records.

2c) In Figs 4 and 5 the authors attempt to link the accumulation records to atmospheric drivers. However, in some regions the simulated accumulation does not match the observed accumulation. Would it be worth repeating these analyses using the modeled accumulation rates for each of the regions? That way you're evaluating something that has consistent internal physics, which would presumably make the correlations stronger and the conceptual picture more clear.

2d) There is much interest in the relationship between temperature and accumulation in Antarctica. Is there a paper planned on this topic within the Antarctic 2k consortium? If not, it would be interesting to include it here. I understand this would go beyond the scope of the present work, and it is of course not a prerequisite for publication.

Minor comments:

P2L11: "composites capture the regional precipitation and SMB variability": I don't understand what this sentence means. Do the composites capture the SMB variability in the models? Or do you see coherence between the records, suggesting a regional signal?

P2L15-16: Do you mean there are only 4 records that cover the last 1000 years, and all 4 show a decrease? Or are there more 1000 year records, but they don't show a decrease? Please clarify

P4L2: I couldn't find Frieler et al in the reference list. I didn't check all references, but there may be more like this. Please check all references.

P5L26-27: I don't get what "equally spaced" means here. The distance between all layers will decrease, whether or not they were equally spaced at the surface. Please rephrase.

P8L9: to clarify: the regional, annual mean from the data are correlated with the annual-mean RACMO values at each grid point?

Fig 2 and 3: is there a way to outline the area of interest? I found myself going back and forth between Figs 1 and 2 to figure out what the acronyms meant again. Alternatively, you could just write out the acronyms in full, in which case we'd know what part of the map is relevant. The figure is not incorrect, it would just be a kind thing to do for the readers.

P8L27: WS appears reasonably well correlated to RACMO over Berkner Island – which is where the only core is from? This could be quantified by extracting the RACMO SMB at the core location, rather than looking at the entire WS area.

P10L6: "high interannual variability": in the data, the model, or both? I suppose sastrugi etc. matter more at low-accumulation sites, as a wind feature of a given size influences more annual layers there?

P12L1: "positive phase of the SAM and the ASL". What is a positive phase of the ASL? Does this mean the ASL has a negative pressure anomaly? Please just write it out in pressure terms.

P12L7: is it possible that the AP snow accumulation anomaly and Bellingshausen sea ice are both just driven by the same SAM trend? Or do you suggest that the

Bellingshausen sea-ice anomaly drives the AP accumulation?

P15L13 What does this "conversion" entail? Is it basically just a linear scaling, or is there more going on? Why not just compare accumulation, rather than SMB? I guess I don't see the added value of this step.

15L14 unitless (no space)

P15L21: Out of the 650 mm/year, the trend of 0.15 mm is just a 0.02% increase. Given the variability and uncertainties (such as in the thinning correction), I would just call this trend zero – i.e. I cannot believe it is statistically robust

P15L31: Is the AP increase the only one that is statistically robust? That would be an important conclusion.

P16L34: Can you express how unusual the current AP trend is in terms of nr. of standard deviations? Is the current trend 2sigma above mean, or 4sigma, for example.

P17L14 Is the SMB increase 44 GT w.e. *per year*? Please check units.

P17L20: I guess for all four regions with a single 1000-year record it is questionable how well a single core represents the entire region – not just WS.

P18L10: Your analysis says nothing about how exceptional the current SMB trend is on the long term perspective. Perhaps you could add an Antarctic-wide histogram to Fig. 8?
* * *

---

## Author Comment (AC2) · 10 Aug 2017

Thank you for your comment and suggestions. I have addressed your concerns and made sure that all the data produced in this study is publicly archived.

Below we address each issue.

Essential additions for this paper: (1) Update the "Data Availability" section to include an explicit URL address to locate the input datasets and the reconstructions generated in this study. For the input datasets, we suggest creating a landing page for this study that lists the 80 individual datasets.

- The data used in this study is stored at the UK Polar Data centre. I've updated the data section with the DOI for the composite data produced by this study.

(2) Add Data Citations (in addition to publication citations) for each of the 80 datasets used in this study. For those data not already in a persistent public repository, submit essential metadata (including the name of the Antarctica region to which it was assigned in this study) along with the accumulation time series itself and add the Data Citation/URL in Table 1.

- I've added a supplementary table to the paper. This expands upon table 1, to include the data citation for each record used. Sadly, not all of the data is available in a public archive but was emailed to me on request from the data owner. In those cases I have cited the paper it was published in, made reference to the region of Antarctica, and provided the contact of the data provider. This information is also stored alongside the composite data stored at the UK Polar Data centre.

- Following discussions with data managers at UK PDC I am not in a position to upload and store data owned by other researchers.

3) If any of the cores used in this study were also used in previous PAGES 2k databases (temperature or isotopes), please include cross references to those IDs.

- None of the records were used in the previous 2k database as they only include isotopes and temperature proxies.

(4) Submit the primary outcome of this study, the composite accumulation time series by region (Fig 7), to a public repository and include the Data Citation/URL in " Data Availability".

- Data submitted to UK PDC and a DOI has been issued.

Recommended elements are:

(5) Archive the gridded spatial correlations of snow accumulation data shown in Figs 2 though 5.

- I have not been able to archive the gridded data. The composite records are freely

available, as are the ECMWF and RACMO data, ensuring that any other researcher can replicate the spatial plots should they wish. I've included a reference to the ECMWF and RACMO data.

(6) Archive the 50- and 100-year trends for each record as shown in Fig 8.

- Again, the original data is available to ensure the trends can be reproduced. I am not sure that uploading a separate document, with just the trends, would be of value. It has taken several weeks to get the DOIs for the UK PDC so I have taken the decision to focus on the essential elements.

---

## Editor Comment (EC1) · C. Turney (Editor) · 21 Aug 2017

Dear Liz,

Just following up on the decision email I sent yesterday. Can I please just reiterate that this special issue is also a contribution to the PAGES Data Stewardship Integrative Activity. As such, all of the papers have been reviewed by the 2k special issue data review team in an effort to attain a high and consistent level of data stewardship across the volume. This involves enacting the publisher's data policy of including a 'data availability' section in each paper, which specifies where the essential data used in the study are located. If extenuating circumstances preclude the public release of data used in a paper, the reason must be clearly stated. In addition, all authors of the

special issue are using data citations so the source of the data can be tracked and attributed to the data generators. I appreciate that the UK Polar Data Center will not accept other people's data via your manuscript. But for any data you are using that is not currently in the public domain can you please reach out to the original authors and get them to do so using a public database (such as NOAA Paleoclimatology). Only then can the results of your study be reproduced by others.

Hope this is all doable. Please let me know if you have any questions. Very happy to discuss further.

With warm wishes,

Chris
* * *

---

## Author Response (AR2)

Author's response

We thank two anonymous reviewers and the PAGES data team for their helpful and constructive comments and suggestions. Thank you for the helpful and constructive suggestions. We have addressed all the concerns raised and made all the minor

5   corrections to the revised text. We have done the additional analysis suggested and feel the paper is much improved as a result.

Below we address each comment individually.

In addition to the specific reviewer comments we have also updated the histograms of 50-year and 100-year trends (Fig. 8). In the previous version the full length of the record was used for each region but now we have selected only the time period

10   1800-2010, where we have greatest coverage. The histograms are now all representing the same time period and highlight some interesting trends and relationships that were missed in the previous interpretation.

Anonymous Referee #1

Overall comments

15   The authors compile some 80 Antarctic ice core records that meet their requirements for temporal coverage, time resolution, and corrections for layer thinning. The records are grouped into regions, composited, and then the regional trends and variability over the past 200-1000 years are discussed. Finally, estimate an overall increase in SMB of ~44 GT since 1800 AD, with much of it occurring within the past couple decades. In general, the paper is very well written and logically organized. It is hard to find a major fault with this paper. It is an accomplishment just to compile the records, requiring the

20   cooperation of scientists from many nations and reflecting many years of field work. If anything, the paper is a bit too guarded and tentative at times: "However, this is just a qualitative explanation, more research using model and field data would be needed to prove this." or "The reduced period of overlap...makes this interpretation less C1 reliable." and many other examples. Caveats are of course a natural part of science, but the inclusion of so many in this paper prevents it from being the final word on snow accumulation or even a paper that would get cited a lot (perhaps they are planning a Nature

25   paper that will pack more punch.).

*In response to the general comments (and following discussions with others) we have decided to shorten the title. Removing the word "review" from the title will hopefully increase the papers impact as this is an independent study into Antarctica mass balance, not just a review.*

30   Specific comments

Affiliations, page 1: I think some of the affiliations are incorrect, please check. For instance, I believe B. Medley is at #9 (NASA), not #10 (U Victoria).

*Updated*

Abstract, line 14: increase in SMB across grounded AIS of ~44 GT since 1800: Some context for this number would be helpful. Is that a lot in terms of mitigating SLR? What is the SLR equivalent? Does this number make sense in terms of published global sea level budgets over the past 100-200 years (is it in the noise or a significant number?)?

*During the revision process we updated the RACMO data (version 2.3p2) and noticed a minor error in the mask we had selected for certain regions (which included ocean as well as land). The data has all be updated and the new values of total SMB change included.*

*The total AIS SMB has been presented in a new figure, with a histogram of the running 50-year and 100-year trends.*

*The increased SMB has been related to SLR equivalents. To add context we relate the net reduction in sea level as a result of the increase in snowfall in Antarctica since 1800 AD to the estimated sea level contribution from the mass loss in the southern Patagonian ice fields.*

Page 4, first paragraph: Given the projected increase in SMB, is it expected to offset overall ice sheet mass loss; is Antarctica expected to be a net contributor to SLR given the overall mass budget?

*The snowfall is expected to mitigate some of the sea level, but not completely offset the ice loss. In the introduction I am reviewing the current literature but make no claims that snowfall will offset mass loss.*

Page 4, line 30: PAGES Antarctica 2K community: Only a select few readers may know what PAGES is, let alone the 2K community. Define?

*Defined PAGES and future earth*

Page 6, lines 11-14: Was their any requirement for proven dating precision and accuracy? Are we assuming that all of the records are perfectly dated?

*A full assessment of all the published age-scales for each ice core was beyond the scope of this study. All data submitted to the Antarctica 2k database was required to submit evidence of independent reference horizons (eg volcanic tie-points) or have evidence in the published literature. For data extracted from other databases, or direct from authors, we checked that independent reference horizons had been used when calculating the age-scale. We cannot be 100% confident that dating errors do not exist, but we have confidence that the published snow accumulation records were dated as accurately as possible.*

*The published dating errors range from 1-3 years for the period 1800-2010, increasing to ~5 years for some sites prior to ~1500 AD. This has been added to section 2.3.*

Page 8, lines 19-20: "predominantly positive phase of the IPO/PDO." There was a major shift in the PDO/IPO in 1998-1990, from positive to negative, affecting more than a third of the 1979-2010 period. This has been shown to impact a number of Antarctic climate trends and it may even be reflected in the recent increase in accumulation in the AP and the decrease in VL. So, I don't think it is accurate to say the the IPO/PDO was predominantly in its positive phase.

*I acknowledge that the IPO has changed sign during the instrumental period (1979-2010), so I have made reference to the changing sign of the IPO during this period.*

Pages 13-14: "The principal teleconnection associated with the Rossby wave propagation from the western tropical Pacific...which originates from the central, tropical C2 Pacific." This sentence is repetitive, as well as contradictory (western Pacific vs central Pacific). A rewrite is needed.

*Sentence re-written (pages 13-14).*

5 Also, in the discussion of the VL and AP composites (sections 3.24 and 3.26), I can't help but notice that the teleconnection patterns of these two regions are roughly opposite in sign. See Figure 4d and 4f. I'm surprised this isn't mentioned somewhere in the paper. 4d resembles the trend pattern associated with the negative PDO, which could have played a role in the recent increase in AP accumulation and decrease in VL precipitation. I also wonder why tropical teleconnections aren't mentioned with respect to AP accumulation.

10 *The discussion on SAM, ENSO and PDO has been expanded in the section relating to AP.*

*Reference to the similarities in VL and AP has been expanded in the VL section.*

Page 15, line 14: change "unit less" to "unitless." Page 15, bottom two paragraphs: As mentioned above, it would be interesting to discuss the opposing accumulation trends in terms of the PDO phase and/or the ASL deepening trend.

*Word changed*

15 Page 17, line 27: Change "were, quality" to "were quality"

*Corrected*

Figures 4 and 5: In contrast to figures 2 and 3, no significance levels are indicated on Figures 4 and 5. Could stippling for significance be added to Figures 4 and 5?

*Figures 4 & 5 stippling added for 95% significance*

20 Figure 5 caption, page 23: should be "correlations...cover" or "correlation...covers."

*Changed*
The PAGES Data Stewardship Integrative Activity seeks to advance best practices for sharing data generated and assembled as part of all PAGES-related activities. As part of this activity, a team of reviewers has been constituted for the "Climate of the Past 2000 years" Special Issue. The data team is reviewing the data handling within each of the CP-Discussion papers in

30 relation to the CP data policy and current best practices. The team has identified essential and recommended additions for each paper, with the goal of achieving a high and consistent level of data stewardship across the 2k Special Issue. We recognize that an additional effort will likely be required to meet the high level of data stewardship envisaged, and we appreciate the dedication and contribution of the authors. This includes the use of Data Citations (see example in

supplement). We ask authors to respond to our comments as part of the regular C1 open interactive discussion. If you have any questions about PAGES Data Stewardship principles, please contact any of us directly.

Best wishes for the success of your paper, 2k Special Issue Data Review Team (Darrell Kaufman, Nerilie Abram, Belen Martrat, Raphael Neukom, Scott St. George) and ex-officio team members (Marie-France Loutre, Lucien von Gunten)

5  Essential additions for this paper:

(1) Update the "Data Availability" section to include an explicit URL address to locate the input datasets and the reconstructions generated in this study. For the input datasets, we suggest creating a landing page for this study that lists the 80 individual datasets.

*The data used in this study is stored at the UK Polar Data centre. I've updated the data section with the DOI for the*
10  *composite data produced by this study.*

(2) Add Data Citations (in addition to publication citations) for each of the 80 datasets used in this study. For those data not already in a persistent public repository, submit essential metadata (including the name of the Antarctica region to which it was assigned in this study) along with the accumulation time series itself and add the Data Citation/URL in Table 1.

*I've added a supplementary table to the paper. This expands upon table 1, to include the data citation for each record used. I*
15  *have cited the paper it was first published in, made reference to the region of Antarctica, provided the contact of the data*
*provider and the link to the data centre. This information is also stored alongside the composite data stored at the UK Polar*
*Data centre.*

3) If any of the cores used in this study were also used in previous PAGES 2k databases (temperature or isotopes), please include cross references to those IDs.

20  *None of the records were used in the previous 2k database as they only include isotopes and temperature proxies.*

(4) Submit the primary outcome of this study, the composite accumulation time series by region (Fig 7), to a public repository and include the Data Citation/URL in "

*Data Availability". Data submitted to UK PDC and a DOI has been issued.*

Recommended elements are:

25  (5) Archive the gridded spatial correlations of snow accumulation data shown in Figs 2 though 5.

*I have not been able to archive the gridded data. The composite records are freely available, as are the ECMWF and*
*RACMO data, ensuring that any other researcher can replicate the spatial plots should they wish. I've included a reference*
*to the ECMWF and RACMO data.*

(6) Archive the 50- and 100-year trends for each record as shown in Fig 8.

30  *Again, the original data is available to ensure the trends can be reproduced. I am not sure that uploading a separate*
*document, with just the trends, would be of value. It has taken several weeks to get the DOIs for the UK PDC so I have taken*
*the decision to focus on the essential elements.*

*In response to the editor's request we have encouraged all data owners to upload their original data sets to a recognised data centre. As this wasn't always possible we have decided to store all original snow accumulation data used in this study at the UK Polar Data Centre, alongside the data citations and the regional composites produced in this study.*

5    *Thomas, E. (2017) "Antarctic regional snow accumulation composites over the past 1000 years" Polar Data Centre, Natural Environment Research Council, UK. https://doi.org/10.5285/c4ecfe25-12f2-453b-ad19-49a19e90ee32*

Anonymous Referee #2

Thomas et al. have compiled available high-resolution ice core accumulation records from Antarctica, and review those within the PAGES 2k framework. They divide the cores into 7 regions to provide a regional perspective on ice accumulation and surface mass balance in Antarctica. The surface mass balance of Antarctica is an important topic of study, with implications for the Antarctic contribution to sea-level rise. Overall the study is relevant, interesting, well-written and clear. I

15   propose some minor corrections and additions for readability, procedural clarity and a more thorough analysis.

General comments:

1)   My first concern is procedural clarity and treatment of uncertainties.

1a) Combining individual records must be done carefully to avoid jumps in the composite at the locations where the number

20   of cores changes. There is some discussion in section 2.6, but it could be expanded. How did you normalize the records? Did you scale the variance, or just the mean? Did all records include the full 1960-1990 reference period?

*I appreciate the concern about combining records and that was why it was important to show the number of records in the plots. The largest number of records occurs between 1800 – 2000, and therefore this period was chosen as a focus, rather*

25   *than attempting to draw too much from the longer time period.*

*The text has been clarified to confirm that all records were normalized, based on the mean and standard deviation during the reference period.*

*Yes, all records cover the reference period.*

1b) How are age uncertainties incorporated? Are all chronologies based on annual layer counts? Section 2.3 merely states they need to have annual resolution, so I assume this is the case. The age uncertainties are of course critical when comparing to RACMO and ERA-interim. What is the typical uncertainty in the age scales, and can this influence the conclusions? Also,

dating uncertainties will introduce accumulation uncertainties, because accumulation is essentially the derivative of the depth-age relationship. So a 5% age uncertainty results in a 5% accumulation uncertainty.

*Yes, there are undoubtedly age uncertainties involved. It is beyond the scope of this study to do independent dating on all the records and therefore we have to assume that since all records have been published and peer-reviewed the records are dated as accurately as possible. As a quality control we ensured that all records were referenced against dating horizons, such as volcanic tie points. I have included a line about dating errors in section 2.3*

*"Published dating errors range from 1-3 years for the period 1800-2010, increasing to ~5 years for some sites prior to ~1500 AD."*

1c) I assume that all RACMO and ERA data are annual means? Were these taken as calendar years? The annual proxies used in the layer count could of course represent another time period (e.g. spring to spring).

*Yes, annual means taken as Jan – December. The accumulation data is also assumed to be summer-summer, approximately Jan – December. However, we acknowledge that with ice core annual layer counting the exact timing of a year is not precise.*

1d) How was the layer thinning correction done? Was the treatment identical for all records? Section 2.2 reviews several methods (Nye, Dansgaard-Johnsen and Roberts), but it's not stated which one is used. Was this done on a case-by-case basis? This does not matter for the inter-annual variability, but this is critical for investigating the long-term trends. Details are needed for the reader to evaluate how reliable the trends are. Please elaborate on this.

*Each record used a different thinning model, depending on which was most appropriate. In order to make it clearer I have added a sentence to section 2.3 to confirm that the published thinning function was used.*

1e) What is the typical uncertainty in the thinning correction, and how does it influence the reconstructed centennial-scale trends?

*The influence of vertical strain rate increases with depth. Therefore, uncertainties in the accumulation rate associated with assumptions about the vertical distribution of the vertical strain rate also increase with the relative depth compared to the local ice thickness. Most of the ice cores in this study are relatively shallow, and therefore uncertainty in the vertical strain rate is unlikely to be a major source of error. For the deeper ice cores, it is difficult, in general, to quantify the influence of the uncertainty in the vertical strain rate. The authors are aware of only one study that compares the accumulation rate*

*calculated using difference vertical strain rate models: Roberts et al (2015) found the influence of the vertical strain rate model to be concentrated at lower frequency, and to be small (less than 4% of the accumulation rate).*

*The following has been added to section 2.2:*

*"Uncertainties in the accumulation rate associated with the vertical strain increase with depth. Most of the ice cores in this study are relatively shallow, and therefore uncertainty in the vertical strain rate is expected to be low. Roberts et al (2015) found the influence of the vertical strain rate model to be concentrated at lower frequency, and to be small (less than 4% of the accumulation rate)."*

2) My second suggestions focus on the analysis of the records

2a) The authors test how representative the records are by comparing them to RACMO and ERA-interim using spatial correlation plots, which is very qualitative and includes some hand-waving. Since RACMO and ERA-interim are gridded products, it should be trivial to extract the SMB time series directly at the core locations – these model time series could then be composited for the exact same time periods and in the same way as the ice core records were composited. This will allow for easy quantitative analysis. For example, how well are the model and data composites correlated? How well does the model capture variability within each region? Etc. This could complement the figures provided.

*In order to expand upon the relationship between modelled SMB and ice core derived snow accumulation we have included a plot of the annual average snow accumulation (1979-2010) at each site, overlain on the SMB from RACMO (Fig 1b). We have also added stars to Fig 1a to highlight ice cores where the correlation between snow accumulation and RACMO SMB is significant at p > 0.05.*

*Following the reviewers suggestion we have extracted the RACMO time series at each ice core location and followed the same method to produce a regional composite. The results were helpful in demonstrating that even if the snow accumulation at each site is 100% certain (for example if we assume RACMO SMB to be the true value), in regions such as East Antarctica the resulting composite would still not represent regional SMB. We simply do not have enough data points (ice cores) to provide the spatial coverage needed.*

*However, in regions where we have a greater number of sites (or better spaced sites), such as West Antarctica and the Antarctic Peninsula, we have more confidence that the regional composite is representative of regional SMB. This is demonstrated by high correlations between the ice core derived regional composite and the RACMO derived regional composite.*

*The findings are presented in a supplementary figure (Fig. S1).*

2b) How well do individual cores in a region represent the regional composite? It would be very easy to figure out in a principal component analysis. The variance explained by the first component will tell you how much of the signal variance is shared between the records.

*We have demonstrated how well individual cores represent the regional snow accumulation, based on the correlations with RACMO.*

*Snow accumulation is highly variable spatially and individual ice core sites represent not only the regional climate signal but also local variability and "noise". The intention of this study was to combine records in a given region with the intention of reducing the signal to noise ratio. The results demonstrate that in high accumulation sites, such as WAIS, the signal to noise is low and thus the composite is representative of regional precipitation variability. However, for low accumulation sites where noise is high (sastrugi etc), the individual sites are poorly represents local SMB and therefore the composite poorly represents regional SMB.*

*Regarding PCA, this approach was taken for a previous study of this kind (Frezzotti et al., 2013), but the results were not useful and therefore it was decided not to run this analysis for this paper. I don't feel this analysis would be possible in the time available and based on the expertise from within the group we don't feel it would improve the paper.*

20   2c) In Figs 4 and 5 the authors attempt to link the accumulation records to atmospheric drivers. However, in some regions the simulated accumulation does not match the observed accumulation. Would it be worth repeating these analyses using the modeled accumulation rates for each of the regions? That way you're evaluating something that has consistent internal physics, which would presumably make the correlations stronger and the conceptual picture more clear.

25   *I have repeated the spatial correlations from Fig 4 & 5 using the modelled SMB from RACMO. The plots are presented as supplementary figures. I have added some text in the section relating to WS, to describe the expected relationship with 850 hPa and sea ice and also included a sentence to demonstrate that the pattern between SMB and 850 hPa is similar when using both modelled (RACMO) and ice core derived SMB.*

30   2d) There is much interest in the relationship between temperature and accumulation in Antarctica. Is there a paper planned on this topic within the Antarctic 2k consortium? If not, it would be interesting to include it here. I understand this would go beyond the scope of the present work, and it is of course not a prerequisite for publication.

*Comparing the snow accumulation records with temperature (or $\delta^{18}O$) is planned as a future activity for the PAGES team.*

Minor comments:

P2L11: "composites capture the regional precipitation and SMB variability": I don't understand what this sentence means. Do the composites capture the SMB variability in the models? Or do you see coherence between the records, suggesting a regional signal?

*As defined by the models*

P2L15-16: Do you mean there are only 4 records that cover the last 1000 years, and all 4 show a decrease? Or are there more 1000 year records, but they don't show a decrease?

*Just four records and they show a decrease. I've added "they" to sentence for clarity.*

Please clarify P4L2: I couldn't find Frieler et al in the reference list. I didn't check all references, but there may be more like this. Please check all references.

*References checked and updated*

P5L26-27: I don't get what "equally spaced" means here. The distance between all layers will decrease, whether or not they were equally spaced at the surface.

*Removed "that were equally spaced".*

Please rephrase. P8L9: to clarify: the regional, annual mean from the data are correlated with the annual-mean RACMO values at each grid point?

*Added*

Fig 2 and 3: is there a way to outline the area of interest? I found myself going back and forth between Figs 1 and 2 to figure out what the acronyms meant again. Alternatively, you could just write out the acronyms in full, in which case we'd know what part of the map is relevant. The figure is not incorrect, it would just be a kind thing to do for the readers.

*I've added an outline to the plots to highlight the region of interest.*

P8L27: WS appears reasonably well correlated to RACMO over Berkner Island – which is where the only core is from? This could be quantified by extracting the RACMO SMB at the core location, rather than looking at the entire WS area.

*We have used the RACMO regional composite for the spatial correlations (Fig 4 & 5), rather than the Berkner ice core data. This demonstrates the "expected" relationships with atmospheric circulation and sea ice.*

P10L6: "high interannual variability": in the data, the model, or both? I suppose sastrugi etc. matter more at low-accumulation sites, as a wind feature of a given size influences more annual layers there?

*Both, but the reference in this paragraph is to the data.*

P12L1: "positive phase of the SAM and the ASL". What is a positive phase of the ASL? Does this mean the ASL has a negative pressure anomaly? Please just write it out in pressure terms.

*Clarified*

P12L7: is it possible that the AP snow accumulation anomaly and Bellingshausen sea ice are both just driven by the same SAM trend? Or do you suggest that the Bellingshausen sea-ice anomaly drives the AP accumulation?

*I think both are being driven by the same (or similar) atmospheric drivers, eg SAM. However, there is evidence that changing sea ice will influence the availability of surface level moisture and therefore the effect on snow accumulation is amplified.*

P15L13 What does this "conversion" entail? Is it basically just a linear scaling, or is there more going on? Why not just compare accumulation, rather than SMB? I guess I don't see the added value of this step.

*Updated text description*

15L14 unitless (no space)

*Corrected*

P15L21: Out of the 650 mm/year, the trend of 0.15 mm is just a 0.02% increase. Given the variability and uncertainties (such as in the thinning correction), I would just call this trend zero – i.e. I cannot believe it is statistically robust

I have changed the text to only include the trends that are statistically significant (p<0.01).

P15L31: Is the AP increase the only one that is statistically robust? That would be an important conclusion.

*The trend in four regions (EAP, WS, DML and AP) is statistically significant at the 99% level. However the AP trend is by far the highest.*

P16L34: Can you express how unusual the current AP trend is in terms of nr. of standard deviations? Is the current trend 2sigma above mean, or 4sigma, for example.

*In the original version we already concluded that "the early 2000s exceeds two standard deviations above the record average for the entire 200-year period."*

P17L14 Is the SMB increase 44 GT w.e. *per year*? Please check units.

*The units have now been updated as Gt yr$^{-1}$.*
*During the revision process we updated the RACMO data (version 2.3p2) and noticed a minor error in the mask we had selected for certain regions (which included ocean as well as land). The data has all be updated and the new values of total SMB change included.*

P17L20: I guess for all four regions with a single 1000-year record it is questionable how well a single core represents the entire region – not just WS.

*Yes, which is why we didn't want to draw too much from these sites and expand to the full ~2k available.*

P18L10: Your analysis says nothing about how exceptional the current SMB trend is on the long term perspective. Perhaps you could add an Antarctic-wide histogram to Fig. 8?
*The total AIS SMB has been presented in a new figure, with a histogram of the running 50-year and 100-year trends. Both the most recent 50-year and 100-year trends are significant and appear unusual in the context of the past 300 years.*
*The total SMB increase has been related to relative change in sea level and % of annual average SMB to add context and perspective.*

[revised manuscript text omitted]